# CONTINUAL LEARNING WITH HYPERNETWORKS

**Johannes von Oswald\*, Christian Henning\*, Benjamin F. Grewe, João Sacramento**
\*Equal contribution

Institute of Neuroinformatics
University of Zürich and ETH Zürich
Zürich, Switzerland
`{voswaldj,henningc,bgrewe,rjoao}@ethz.ch`

## ABSTRACT

Artificial neural networks suffer from catastrophic forgetting when they are se-quentially trained on multiple tasks. To overcome this problem, we present a novel approach based on task-conditioned hypernetworks, i.e., networks that generate the weights of a target model based on task identity. Continual learning (CL) is less difficult for this class of models thanks to a simple key feature: instead of recalling the input-output relations of all previously seen data, task-conditioned hypernetworks only require rehearsing task-specific weight realizations, which can be maintained in memory using a simple regularizer. Besides achieving state-of-the-art performance on standard CL benchmarks, additional experiments on long task sequences reveal that task-conditioned hypernetworks display a very large capacity to retain previous memories. Notably, such long memory lifetimes are achieved in a compressive regime, when the number of trainable hypernetwork weights is comparable or smaller than target network size. We provide insight into the structure of low-dimensional task embedding spaces (the input space of the hypernetwork) and show that task-conditioned hypernetworks demonstrate transfer learning. Finally, forward information transfer is further supported by empirical results on a challenging CL benchmark based on the CIFAR-10/100 image datasets.

## 1 INTRODUCTION

We assume that a neural network $f(\mathbf{x}, \Theta)$ with trainable weights $\Theta$ is given data from a set of tasks $\{(\mathbf{X}^{(1)}, \mathbf{Y}^{(1)}), \ldots, (\mathbf{X}^{(T)}, \mathbf{Y}^{(T)})\}$, with input samples $\mathbf{X}^{(t)} = \{\mathbf{x}^{(t,i)}\}_{i=1}^{n_t}$ and output samples $\mathbf{Y}^{(t)} = \{\mathbf{y}^{(t,i)}\}_{i=1}^{n_t}$, where $n_t \equiv |\mathbf{X}^{(t)}|$. A standard training approach learns the model using data from all tasks at once. However, this is not always possible in real-world problems, nor desirable in an online learning setting. Continual learning (CL) refers to an online learning setup in which tasks are presented sequentially (see van de Ven & Tolias, 2019, for a recent review on CL). In CL, when learning a new task $t$, starting with weights $\Theta^{(t-1)}$ and observing only $(\mathbf{X}^{(t)}, \mathbf{Y}^{(t)})$, the goal is to find a new set of parameters $\Theta^{(t)}$ that (1) retains (no catastrophic forgetting) or (2) improves (positive backward transfer) performance on previous tasks compared to $\Theta^{(t-1)}$ and (3) solves the new task $t$ potentially utilizing previously acquired knowledge (positive forward transfer). Achieving these goals is non-trivial, and a longstanding issue in neural networks research.

Here, we propose addressing catastrophic forgetting at the meta level: instead of directly attempting to retain $f(\mathbf{x}, \Theta)$ for previous tasks, we fix the outputs of a metamodel $f_{\mathrm{h}}(\mathbf{e}, \Theta_{\mathrm{h}})$ termed *task-conditioned hypernetwork* which maps a task embedding $\mathbf{e}$ to weights $\Theta$. Now, a *single* point has to be memorized per task. To motivate such approach, we perform a thought experiment: we assume that we are allowed to store all inputs $\{\mathbf{X}^{(1)}, \ldots, \mathbf{X}^{(T)}\}$ seen so far, and to use these data to compute model outputs corresponding to $\Theta^{(T-1)}$. In this idealized setting, one can avoid forgetting by simply mixing data from the current task with data from the past, $\{(\mathbf{X}^{(1)}, \hat{\mathbf{Y}}^{(1)}), \ldots, (\mathbf{X}^{(T-1)}, \hat{\mathbf{Y}}^{(T-1)}), (\mathbf{X}^{(T)}, \mathbf{Y}^{(T)})\}$, where $\hat{\mathbf{Y}}^{(t)}$ refers to a set of synthetic targets generated using the model itself $f(\cdot, \Theta^{(t-1)})$. Hence, by training to retain previously acquired input-output mappings, one can obtain a sequential algorithm in principle as powerful as multi-task learning. Multi-task learning, where all tasks are learned

simultaneously, can be seen as a CL upper-bound. The strategy described above has been termed rehearsal (Robins, 1995). However, storing previous task data violates our CL desiderata.

Therefore, we introduce a change in perspective and move from the challenge of maintaining individual input-output data points to the problem of maintaining sets of parameters $\{\Theta^{(t)}\}$, without explicitly storing them. To achieve this, we train the metamodel parameters $\Theta_h$ analogous to the above outlined learning scheme, where synthetic targets now correspond to weight configurations that are suitable for previous tasks. This exchanges the storage of an entire dataset by a single low-dimensional task descriptor, yielding a massive memory saving in all but the simplest of tasks. Despite relying on regularization, our approach is a conceptual departure from previous algorithms based on regularization in weight (e.g., Kirkpatrick et al., 2017; Zenke et al., 2017) or activation space (e.g., He & Jaeger, 2018).

Our experimental results show that task-conditioned hypernetworks do not suffer from catastrophic forgetting on a set of standard CL benchmarks. Remarkably, they are capable of retaining memories with practically no decrease in performance, when presented with very long sequences of tasks. Thanks to the expressive power of neural networks, task-conditioned hypernetworks exploit task-to-task similarities and transfer information forward in time to future tasks. Finally, the task-conditional metamodelling perspective that we put forth is generic, as it does not depend on the specifics of the target network architecture. We exploit this key principle and show that the very same metamodelling framework extends to, and can improve, an important class of CL methods known as generative replay methods, which are current state-of-the-art performers in many practical problems (Shin et al., 2017; Wu et al., 2018; van de Ven & Tolias, 2018).

## 2 MODEL

### 2.1 TASK-CONDITIONED HYPERNETWORKS

**Hypernetworks parameterize target models.** The centerpiece of our approach to continual learning is the hypernetwork, Fig. 1a. Instead of learning the parameters $\Theta_{trgt}$ of a particular function $f_{trgt}$ directly (the *target model*), we learn the parameters $\Theta_h$ of a metamodel. The output of such metamodel, the hypernetwork, is $\Theta_{trgt}$. Hypernetworks can therefore be thought of as weight generators, which were originally introduced to dynamically parameterize models in a compressed form (Ha et al., 2017; Schmidhuber, 1992; Bertinetto et al., 2016; Jia et al., 2016).

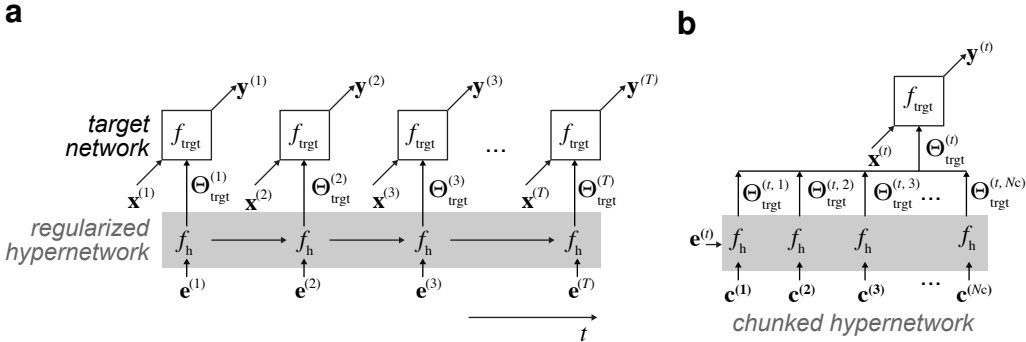

Figure 1: **Task-conditioned hypernetworks for continual learning. (a)** Commonly, the parameters of a neural network are directly adjusted from data to solve a task. Here, a weight generator termed *hypernetwork* is learned instead. Hypernetworks map embedding vectors to weights, which parameterize a target neural network. In a continual learning scenario, a set of task-specific embeddings is learned via backpropagation. Embedding vectors provide task-dependent context and bias the hypernetwork to particular solutions. **(b)** A smaller, chunked hypernetwork can be used iteratively, producing a chunk of target network weights at a time (e.g., one layer at a time). Chunked hypernetworks can achieve model compression: the effective number of trainable parameters can be smaller than the number of target network weights.

**Continual learning with hypernetwork output regularization.** One approach to avoid catastrophic forgetting is to store data from previous tasks and corresponding model outputs, and then fix such outputs. This can be achieved using an output regularizer of the following form, where past outputs play the role of pseudo-targets (Robins, 1995; Li & Hoiem, 2018; Benjamin et al., 2018):

$$\mathcal{L}_{\text{output}} = \sum_{t=1}^{T-1} \sum_{i=1}^{|\mathbf{X}^{(t)}|} \| f(\mathbf{x}^{(t,i)}, \Theta^*) - f(\mathbf{x}^{(t,i)}, \Theta) \|^2, \tag{1}$$

In the equation above, $\Theta^*$ is the set of parameters before attempting to learn task $T$, and $f$ is the learner. This approach, however, requires storing and iterating over previous data, a process that is known as *rehearsing*. This is potentially expensive memory-wise and not strictly online learning. A possible workaround is to generate the pseudo-targets by evaluating $f$ on random patterns (Robins, 1995) or on the current task dataset (Li & Hoiem, 2018). However, this does not necessarily fix the behavior of the function $f$ in the regions of interest.

Hypernetworks sidestep this problem naturally. In target network weight space, a *single* point (i.e., one set of weights) has to be fixed per task. This can be efficiently achieved with task-conditioned hypernetworks, by fixing the hypernetwork output on the appropriate task embedding.

Similar to Benjamin et al. (2018), we use a two-step optimization procedure to introduce memory-preserving hypernetwork output constraints. First, we compute a candidate change $\Delta\Theta_{\text{h}}$ which minimizes the current task loss $\mathcal{L}_{\text{task}}^{(T)} = \mathcal{L}_{\text{task}}(\Theta_{\text{h}}, \mathbf{e}^{(T)}, \mathbf{X}^{(T)}, \mathbf{Y}^{(T)})$ with respect to $\Theta$. The candidate $\Delta\Theta_{\text{h}}$ is obtained with an optimizer of choice (we use Adam throughout; Kingma & Ba, 2015). The actual parameter change is then computed by minimizing the following total loss:

$$\mathcal{L}_{\text{total}} = \mathcal{L}_{\text{task}}(\Theta_{\text{h}}, \mathbf{e}^{(T)}, \mathbf{X}^{(T)}, \mathbf{Y}^{(T)}) + \mathcal{L}_{\text{output}}(\Theta_{\text{h}}^*, \Theta_{\text{h}}, \Delta\Theta_{\text{h}}, \{\mathbf{e}^{(t)}\})$$

$$= \mathcal{L}_{\text{task}}(\Theta_{\text{h}}, \mathbf{e}^{(T)}, \mathbf{X}^{(T)}, \mathbf{Y}^{(T)}) + \frac{\beta_{\text{output}}}{T-1} \sum_{t=1}^{T-1} \| f_{\text{h}}(\mathbf{e}^{(t)}, \Theta_{\text{h}}^*) - f_{\text{h}}(\mathbf{e}^{(t)}, \Theta_{\text{h}} + \Delta\Theta_{\text{h}})) \|^2, \tag{2}$$

where $\Theta_{\text{h}}^*$ is the set of hypernetwork parameters before attempting to learn task $T$, $\Delta\Theta_{\text{h}}$ is considered fixed and $\beta_{\text{output}}$ is a hyperparameter that controls the strength of the regularizer. On Appendix D, we run a sensitivity analysis on $\beta_{\text{output}}$ and experiment with a more efficient stochastic regularizer where the averaging is performed over a random subset of past tasks.

More computationally-intensive algorithms that involve a full inner-loop refinement, or use second-order gradient information by backpropagating through $\Delta\Theta_{\text{h}}$ could be applied. However, we found empirically that our one-step correction worked well. Exploratory hyperparameter scans revealed that the inclusion of the lookahead $\Delta\Theta_h$ in (2) brought a minor increase in performance, even when computed with a cheap one-step procedure. Note that unlike in Eq. 1, the memory-preserving term $\mathcal{L}_{\text{output}}$ does not depend on past data. Memory of previous tasks enters only through the collection of task embeddings $\{\mathbf{e}^{(t)}\}_{t=1}^{T-1}$.

**Learned task embeddings.** Task embeddings are differentiable deterministic parameters that can be learned, just like $\Theta_{\text{h}}$. At every learning step of our algorithm, we also update the current task embedding $\mathbf{e}^{(T)}$ to minimize the task loss $\mathcal{L}_{\text{task}}^{(T)}$. After learning the task, the final embedding is saved and added to the collection $\{\mathbf{e}^{(t)}\}$.

## 2.2 Model compression with chunked hypernetworks

**Chunking.** In a straightforward implementation, a hypernetwork produces the entire set of weights of a target neural network. For modern deep neural networks, this is a very high-dimensional output. However, hypernetworks can be invoked iteratively, filling in only part of the target model at each step, in *chunks* (Ha et al., 2017; Pawlowski et al., 2017). This strategy allows applying smaller hypernetworks that are reusable. Interestingly, with chunked hypernetworks it is possible to solve tasks in a compressive regime, where the number of learned parameters (those of the hypernetwork) is effectively smaller than the number of target network parameters.

**Chunk embeddings and network partitioning.** Reapplying the same hypernetwork multiple times introduces weight sharing across partitions of the target network, which is usually not desirable.

To allow for a flexible parameterization of the target network, we introduce a set $\mathcal{C} = \{\mathbf{c}_i\}_{i=1}^{N_\mathrm{C}}$ of chunk embeddings, which are used as an additional input to the hypernetwork, Fig. 1b. Thus, the full set of target network parameters $\Theta_\mathrm{trgt} = [f_\mathrm{h}(\mathbf{e}, \mathbf{c}_1), \ldots, f_\mathrm{h}(\mathbf{e}, \mathbf{c}_{N_\mathrm{C}})]$ is produced by iteration over $\mathcal{C}$, keeping the task embedding $\mathbf{e}$ fixed. This way, the hypernetwork can produce distinct weights for each chunk. Furthermore, chunk embeddings, just like task embeddings, are ordinary deterministic parameters that we learn via backpropagation. For simplicity, we use a shared set of chunk embeddings for all tasks and we do not explore special target network partitioning strategies.

How flexible is our approach? Chunked neural networks can in principle approximate any target weight configuration arbitrarily well. For completeness, we state this formally in Appendix E.

## 2.3 CONTEXT-FREE INFERENCE: UNKNOWN TASK IDENTITY

**Determining which task to solve from input data.** Our hypernetwork requires a task embedding input to generate target model weights. In certain CL applications, an appropriate embedding can be immediately selected as task identity is unambiguous, or can be readily inferred from contextual clues. In other cases, knowledge of the task at hand is not explicitly available during inference. In the following, we show that our metamodelling framework generalizes to such situations. In particular, we consider the problem of inferring which task to solve from a given input pattern, a noted benchmark challenge (Farquhar & Gal, 2018; van de Ven & Tolias, 2019). Below, we explore two different strategies that leverage task-conditioned hypernetworks in this CL setting.

**Task-dependent predictive uncertainty.** Neural network models are increasingly reliable in signalling novelty and appropriately handling out-of-distribution data. For categorical target distributions, the network ideally produces a flat, high entropy output for unseen data and, conversely, a peaked, low-entropy response for in-distribution data (Hendrycks & Gimpel, 2016; Liang et al., 2017). This suggests a first, simple method for task inference (HNET+ENT). Given an input pattern for which task identity is unknown, we pick the task embedding which yields lowest predictive uncertainty, as quantified by output distribution entropy. While this method relies on accurate novelty detection, which is in itself a far from solved research problem, it is otherwise straightforward to implement and no additional learning or model is required to infer task identity.

**Hypernetwork-protected synthetic replay.** When a generative model is available, catastrophic forgetting can be circumvented by mixing current task data with replayed past synthetic data (for recent work see Shin et al., 2017; Wu et al., 2018). Besides protecting the generative model itself, synthetic data can protect another model of interest, for example, another discriminative model. This conceptually simple strategy is in practice often the state-of-the-art solution to CL (van de Ven & Tolias, 2019). Inspired by these successes, we explore augmenting our system with a replay network, here a standard variational autoencoder (VAE; Kingma & Welling, 2014) (but see Appendix F for experiments with a generative adversarial network, Goodfellow et al., 2014).

Synthetic replay is a strong, but not perfect, CL mechanism as the generative model is subject to drift, and errors tend to accumulate and amplify with time. Here, we build upon the following key observation: just like the target network, the generator of the replay model can be specified by a hypernetwork. This allows protecting it with the output regularizer, Eq. 2, rather than with the model's own replay data, as done in related work. Thus, in this combined approach, both synthetic replay and task-conditional metamodelling act in tandem to reduce forgetting.

We explore hypernetwork-protected replay in two distinct setups. First, we consider a minimalist architecture (HNET+R), where *only the replay model*, and not the target classifier, is parameterized by a hypernetwork. Here, forgetting in the target network is obviated by mixing current data with synthetic data. Synthetic target output values for previous tasks are generated using a soft targets method, i.e., by simply evaluating the target function before learning the new task on synthetic input data. Second (HNET+TIR), we introduce an auxiliary task inference classifier, protected using synthetic replay data and trained to predict task identity from input patterns. This architecture requires additional modelling, but it is likely to work well when tasks are strongly dissimilar. Furthermore, the task inference subsystem can be readily applied to process more general forms of contextual information, beyond the current input pattern. We provide additional details, including network architectures and the loss functions that are optimized, in Appendices B and C.

## 3 RESULTS

We evaluate our method on a set of standard image classification benchmarks on the MNIST, CIFAR-10 and CIFAR-100 public datasets[1]. Our main aims are to (1) study the memory retention capabilities of task-conditioned hypernetworks across three continual learning settings, and (2) investigate information transfer across tasks that are learned sequentially.

**Continual learning scenarios.** In our experiments we consider three different CL scenarios (van de Ven & Tolias, 2019). In CL1, the task identity is given to the system. This is arguably the standard sequential learning scenario, and the one we consider unless noted otherwise. In CL2, task identity is unknown to the system, but it does not need to be explicitly determined. A target network with a fixed head is required to solve multiple tasks. In CL3, task identity has to be explicitly inferred. It has been argued that this scenario is the most natural, and the one that tends to be harder for neural networks (Farquhar & Gal, 2018; van de Ven & Tolias, 2019).

**Experimental details.** Aiming at comparability, for the experiments on the MNIST dataset we model the target network as a fully-connected network and set all hyperparameters after van de Ven & Tolias (2019), who recently reviewed and compared a large set of CL algorithms. For our CIFAR experiments, we opt for a ResNet-32 target neural network (He et al., 2016) to assess the scalability of our method. A summary description of the architectures and particular hyperparameter choices, as well as additional experimental details, is provided in Appendix C. We emphasize that, on all our experiments, the number of hypernetwork parameters is always smaller or equal than the number of parameters of the models we compare with.

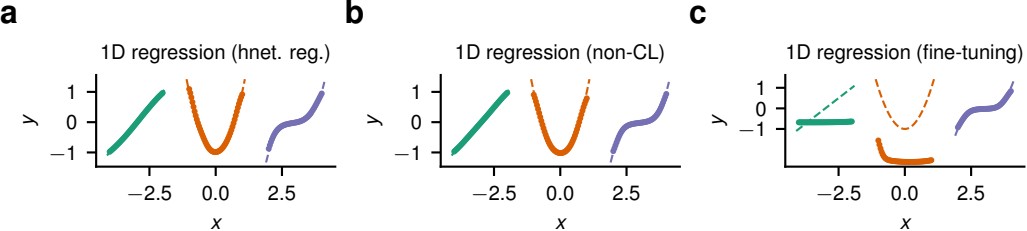

Figure 2: **1D nonlinear regression. (a)** Task-conditioned hypernetworks with output regularization can easily model a sequence of polynomials of increasing degree, while learning in a continual fashion. **(b)** The solution found by a target network which is trained directly on all tasks simultaneously is similar. **(c)** Fine-tuning, i.e., learning sequentially, leads to forgetting of past tasks. Dashed lines depict ground truth, markers show model predictions.

**Nonlinear regression toy problem.** To illustrate our approach, we first consider a simple nonlinear regression problem, where the function to be approximated is scalar-valued, Fig. 2. Here, a sequence of polynomial functions of increasing degree has to be inferred from noisy data. This motivates the continual learning problem: when learning each task in succession by modifying $\Theta_h$ with the memory-preserving regularizer turned off ($\beta_{output} = 0$, see Eq. 2) the network learns the last task but forgets previous ones, Fig. 2c. The regularizer protects old solutions, Fig. 2a, and performance is comparable to an offline non-continual learner, Fig. 2b.

**Permuted MNIST benchmark.** Next, we study the permuted MNIST benchmark. This problem is set as follows. First, the learner is presented with the full MNIST dataset. Subsequently, novel tasks are obtained by applying a random permutation to the input image pixels. This process can be repeated to yield a long task sequence, with a typical length of $T = 10$ tasks. Given the low similarity of the generated tasks, permuted MNIST is well suited to study the memory capacity of a continual learner. For $T = 10$, we find that task-conditioned hypernetworks are state-of-the-art on CL1, Table 1. Interestingly, inferring tasks through the predictive distribution entropy (HNET+ENT) works well on the permuted MNIST benchmark. Despite the simplicity of the method, both synaptic intelligence (SI; Zenke et al., 2017) and online elastic weight consolidation (EWC; Schwarz et al., 2018) are overperformed on CL3 by a large margin. When complemented with generative replay

---

[1]Source code is available under https://github.com/chrhenning/hypercl.

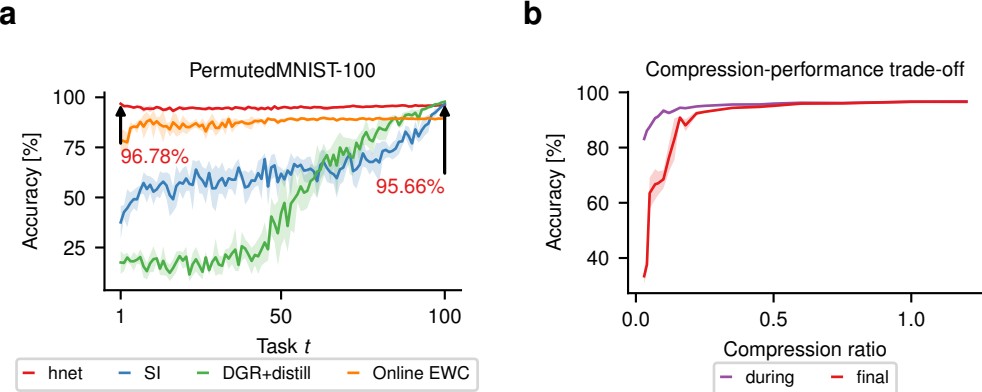

Figure 3: **Experiments on the permuted MNIST benchmark.** (a) Final test set classification accuracy on the $t$-th task after learning one hundred permutations (PermutedMNIST-100). Task-conditioned hypernetworks (hnet, in red) achieve very large memory lifetimes on the permuted MNIST benchmark. Synaptic intelligence (SI, in blue; Zenke et al., 2017), online EWC (in orange; Schwarz et al., 2018) and deep generative replay (DGR+distill, in green; Shin et al., 2017) methods are shown for comparison. Memory retention in SI and DGR+distill degrade gracefully, whereas EWC suffers from rigidity and can never reach very high accuracy, even though memories persist for the entire experiment duration. (b) Compression ratio $\frac{|\Theta_{\mathrm{h}} \cup \{\mathbf{e}^{(t)}\}|}{|\Theta_{\mathrm{trgt}}|}$ versus task-averaged test set accuracy after learning all tasks (labelled 'final', in red) and immediately after learning a task (labelled 'during', in purple) for the PermutedMNIST-10 benchmark. Hypernetworks allow for model compression and perform well even when the number of target model parameters exceeds their own. Performance decays nonlinearly: accuracies stay approximately constant for a wide range of compression ratios below unity. Hyperparameters were tuned once for compression ratio $\approx 1$ and were then used for all compression ratios. Shaded areas denote STD (a) resp. SEM (b) across 5 random seeds.

methods, task-conditioned hypernetworks (HNET+TIR and HNET+R) are the best performers on all three CL scenarios.

Performance differences become larger in the long sequence limit, Fig. 3a. For longer task sequences ($T = 100$), SI and DGR+distill (Shin et al., 2017; van de Ven & Tolias, 2018) degrade gracefully, while the regularization strength of online EWC prevents the method from achieving high accuracy (see Fig. A6 for a hyperparameter search on related work). Notably, task-conditioned hypernetworks show minimal memory decay and find high performance solutions. Because the hypernetwork operates in a compressive regime (see Fig. 3b and Fig. A7 for an exploration of compression ratios), our results do not naively rely on an increase in the number of parameters. Rather, they suggest that previous methods are not yet capable of making full use of target model capacity in a CL setting. We report a set of extended results on this benchmark on Appendix D, including a study of CL2/3 ($T = 100$), where HNET+TIR strongly outperforms the related work.

**Split MNIST benchmark.** Split MNIST is another popular CL benchmark, designed to introduce task overlap. In this problem, the various digits are sequentially paired and used to form five binary classification tasks. Here, we find that task-conditioned hypernetworks are the best overall performers. In particular, HNET+R improves the previous state-of-the-art method DGR+distill on both CL2 and CL3, almost saturating the CL2 upper bound for replay models (Appendix D). Since HNET+R is essentially hypernetwork-protected DGR, these results demonstrate the generality of task-conditioned hypernetworks as effective memory protectors. To further support this, in Appendix F we show that our replay models (we experiment with both a VAE and a GAN) can learn in a class-incremental manner the full MNIST dataset. Finally, HNET+ENT again outperforms both EWC and SI, without any generative modelling.

On the split MNIST problem, tasks overlap and therefore continual learners can transfer information across tasks. To analyze such effects, we study task-conditioned hypernetworks with two-dimensional task embedding spaces, which can be easily visualized. Despite learning happening continually, we

Table 1: Task-averaged test accuracy ($\pm$ SEM, $n = 20$) on the permuted ('P10') and split ('S') MNIST experiments. In the table, EWC refers to online EWC and DGR refers to DGR+distill (results reproduced from van de Ven & Tolias, 2019). We tested three hypernetwork-based models: for HNET+ENT (HNET alone for `CL1`), we inferred task identity based on the entropy of the predictive distribution; for HNET+TIR, we trained a hypernetwork-protected recognition-replay network (based on a VAE, cf. Fig. A1) to infer the task from input patterns; for HNET+R the main classifier was trained by mixing current task data with synthetic data generated from a hypernetwork-protected VAE.

| | EWC | SI | DGR | HNET+ENT | HNET+TIR | HNET+R |
|---|---|---|---|---|---|---|
| P10-`CL1` | $95.96 \pm 0.06$ | $94.75 \pm 0.14$ | $97.51 \pm 0.01$ | $97.57 \pm 0.02$ | $97.57 \pm 0.02$ | $\mathbf{97.87 \pm 0.01}$ |
| P10-`CL2` | $94.42 \pm 0.13$ | $95.33 \pm 0.11$ | $97.35 \pm 0.02$ | $92.80 \pm 0.15$ | $97.58 \pm 0.02$ | $\mathbf{97.60 \pm 0.01}$ |
| P10-`CL3` | $33.88 \pm 0.49$ | $29.31 \pm 0.62$ | $96.38 \pm 0.03$ | $91.75 \pm 0.21$ | $97.59 \pm 0.01$ | $\mathbf{97.76 \pm 0.01}$ |
| S-`CL1` | $99.12 \pm 0.11$ | $99.09 \pm 0.15$ | $99.61 \pm 0.02$ | $99.79 \pm 0.01$ | $99.79 \pm 0.01$ | $\mathbf{99.83 \pm 0.01}$ |
| S-`CL2` | $64.32 \pm 1.90$ | $65.36 \pm 1.57$ | $96.83 \pm 0.20$ | $87.01 \pm 0.47$ | $94.43 \pm 0.28$ | $\mathbf{98.00 \pm 0.03}$ |
| S-`CL3` | $19.96 \pm 0.07$ | $19.99 \pm 0.06$ | $91.79 \pm 0.32$ | $69.48 \pm 0.80$ | $89.59 \pm 0.59$ | $\mathbf{95.30 \pm 0.13}$ |

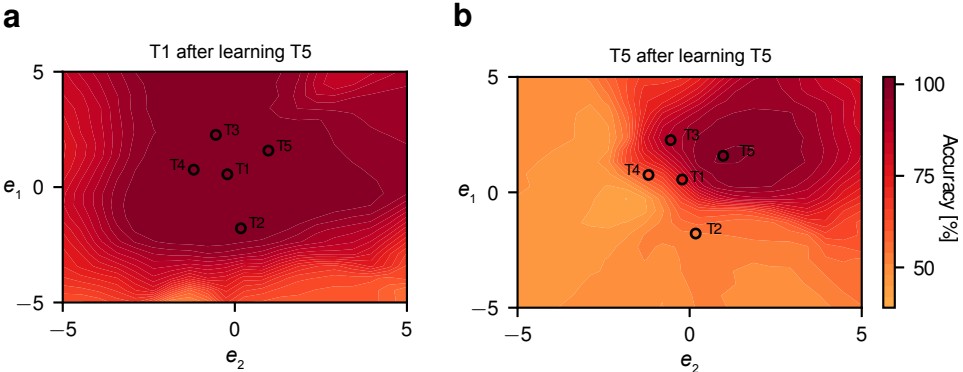

Figure 4: **Two-dimensional task embedding space for the split MNIST benchmark.** Color-coded test set classification accuracies after learning the five splits, shown as the embedding vector components are varied. Markers denote the position of final task embeddings. **(a)** High classification performance with virtually no forgetting is achieved even when **e**-space is low-dimensional. The model shows information transfer in embedding space: the first task is solved in a large volume that includes embeddings for subsequently learned tasks. **(b)** Competition in embedding space: the last task occupies a finite high performance region, with graceful degradation away from the embedding vector. Previously learned task embeddings still lead to moderate, above-chance performance.

find that the algorithm converges to a hypernetwork configuration that can produce target model parameters that simultaneously solve old and new tasks, Fig. 4, given the appropriate task embedding.

**Split CIFAR-10/100 benchmark.** Finally, we study a more challenging benchmark, where the learner is first asked to solve the full CIFAR-10 classification task and is then presented with sets of ten classes from the CIFAR-100 dataset. We perform experiments both with a high-performance ResNet-32 target network architecture (Fig. 5) and with a shallower model (Fig. A3) that we exactly reproduced from previous work (Zenke et al., 2017). Remarkably, on the ResNet-32 model, we find that task-conditioned hypernetworks essentially eliminate altogether forgetting. Furthermore, forward information transfer takes place; knowledge from previous tasks allows the network to find better solutions than when learning each task individually from initial conditions. Interestingly, forward transfer is stronger on the shallow model experiments (Fig. A3), where we otherwise find that our method performs comparably to SI.

## 4 DISCUSSION

**Bayesian accounts of continual learning.** According to the standard Bayesian CL perspective, a posterior parameter distribution is recursively updated using Bayes' rule as tasks arrive

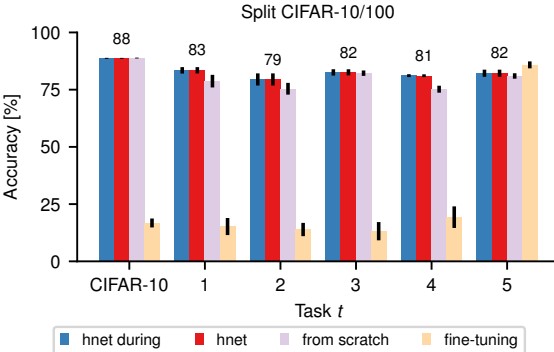

Figure 5: **Split CIFAR-10/100 CL benchmark.** Test set accuracies (mean ± STD, $n = 5$) on the entire CIFAR-10 dataset and subsequent CIFAR-100 splits of ten classes. Our hypernetwork-protected ResNet-32 displays virtually no forgetting; final averaged performance (hnet, in red) matches the immediate one (hnet-during, in blue). Furthermore, information is transferred across tasks, as performance is higher than when training each task from scratch (purple). Disabling our regularizer leads to strong forgetting (in yellow).

(Kirkpatrick et al., 2017; Huszár, 2018; Nguyen et al., 2018). While this approach is theoretically sound, in practice, the approximate inference methods that are typically preferred can lead to stiff models, as a compromise solution that suits all tasks has to be found within the mode determined by the first task. Such restriction does not apply to hypernetworks, which can in principle model complex multimodal distributions (Louizos & Welling, 2017; Pawlowski et al., 2017; Henning et al., 2018). Thus, rich, hypernetwork-modelled priors are one avenue of improvement for Bayesian CL methods. Interestingly, *task-conditioning* offers an alternative possibility: instead of consolidating every task onto a single distribution, a shared task-conditioned hypernetwork could be leveraged to model a set of parameter posterior distributions. This conditional metamodel naturally extends our framework to the Bayesian learning setting. Such approach will likely benefit from additional flexibility, compared to conventional recursive Bayesian updating.

**Related approaches that rely on task-conditioning.** Our model fits within, and in certain ways generalizes, previous CL methods that condition network computation on task descriptors. Task-conditioning is commonly implemented using multiplicative masks at the level of modules (Rusu et al., 2016; Fernando et al., 2017), neurons (Serra et al., 2018; Masse et al., 2018) or weights (Mallya & Lazebnik, 2018). Such methods work best with large networks and come with a significant storage overhead, which typically scales with the number of tasks. Our approach differs by explicitly modelling the full parameter space using a metamodel, the hypernetwork. Thanks to this metamodel, generalization in parameter and task space is possible, and task-to-task dependencies can be exploited to efficiently represent solutions and transfer present knowledge to future problems. Interestingly, similar arguments have been drawn in work developed concurrently to ours (Lampinen & McClelland, 2019), where task embedding spaces are further explored in the context of few-shot learning. In the same vein, and like the approach developed here, recent work in CL generates last-layer network parameters as part of a pipeline to avoid catastrophic forgetting (Hu et al., 2019) or distills parameters onto a contractive auto-encoding model (Camp et al., 2018).

**Positive backwards transfer.** In its current form, the hypernetwork output regularizer protects previously learned solutions from changing, such that only weak backwards transfer of information can occur. Given the role of selective forgetting and refinement of past memories in achieving intelligent behavior (Brea et al., 2014; Richards & Frankland, 2017), investigating and improving backwards transfer stands as an important direction for future research.

**Relevance to systems neuroscience.** Uncovering the mechanisms that support continual learning in both brains and artificial neural networks is a long-standing question (McCloskey & Cohen, 1989; French, 1999; Parisi et al., 2019). We close with a speculative systems interpretation (Kumaran et al., 2016; Hassabis et al., 2017) of our work as a model for modulatory top-down signals in cortex. Task embeddings can be seen as low-dimensional context switches, which determine the behavior of a modulatory system, the hypernetwork in our case. According to our model, the hypernetwork would in turn regulate the activity of a target cortical network.

As it stands, implementing a hypernetwork would entail dynamically changing the entire connectivity of a target network, or cortical area. Such a process seems difficult to conceive in the brain. However, this strict literal interpretation can be relaxed. For example, a hypernetwork can output lower-dimensional modulatory signals (Marder, 2012), instead of a full set of weights. This interpretation

is consistent with a growing body of work which suggests the involvement of modulatory inputs in implementing context- or task-dependent network mode-switching (Mante et al., 2013; Jaeger, 2014; Stroud et al., 2018; Masse et al., 2018).

## 5 CONCLUSION

We introduced a novel neural network model, the task-conditioned hypernetwork, that is well-suited for CL problems. A task-conditioned hypernetwork is a metamodel that learns to parameterize target functions, that are specified and identified in a compressed form using a task embedding vector. Past tasks are kept in memory using a hypernetwork output regularizer, which penalizes changes in previously found target weight configurations. This approach is scalable and generic, being applicable as a standalone CL method or in combination with generative replay. Our results are state-of-the-art on standard benchmarks and suggest that task-conditioned hypernetworks can achieve long memory lifetimes, as well as transfer information to future tasks, two essential properties of a continual learner.

### ACKNOWLEDGMENTS

This work was supported by the Swiss National Science Foundation (B.F.G. CRSII5-173721), ETH project funding (B.F.G. ETH-20 19-01) and funding from the Swiss Data Science Center (B.F.G, C17-18, J. v. O. P18-03). Special thanks to Simone Carlo Surace, Adrian Huber, Xu He, Markus Marks, Maria R. Cervera and Jannes Jegminat for discussions, helpful pointers to the CL literature and for feedback on our paper draft.

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

## A  TASK-CONDITIONED HYPERNETWORKS: MODEL SUMMARY

In our model, a task-conditioned hypernetwork produces the parameters $\Theta_{\text{trgt}} = f_{\text{h}}(\mathbf{e}, \Theta_{\text{h}})$ of a target neural network. Given one such parameterization, the target model then computes predictions $\hat{\mathbf{y}} = f_{\text{trgt}}(\mathbf{x}, \Theta_{\text{trgt}})$ based on input data. Learning amounts to adapting the parameters $\Theta_{\text{h}}$ of the hypernetwork, including a set of task embeddings $\{\mathbf{e}^{(t)}\}_{t=1}^{T}$, as well as a set of chunk embeddings $\{\mathbf{c}_i\}_{i=1}^{N_{\text{C}}}$ in case compression is sought or if the full hypernetwork is too large to be handled directly. To avoid castastrophic forgetting, we introduce an output regularizer which fixes the behavior of the hypernetwork by penalizing changes in target model parameters that are produced for previously learned tasks.

**Variables that need to be stored while learning new tasks.** What are the storage requirements of our model, when learning continually?

1. Memory retention relies on saving one embedding per task. This collection $\{\mathbf{e}^{(t)}\}_{t=1}^{T}$ therefore grows linearly with $T$. Such linear scaling is undesirable asymptotically, but it turns out to be essentially negligible in practice, as each embedding is a single low-dimensional vector (e.g., see Fig. 4 for a run with 2D embeddings).

2. A frozen snapshot of the hypernetwork parameters $\Theta_{\mathrm{h}}^{*}$, taken before learning a new task, needs to be kept, to evaluate the output regularizer in Eq. 2.

## B  ADDITIONAL DETAILS ON HYPERNETWORK-PROTECTED REPLAY MODELS

**Variational autoencoders.** For all HNET+TIR and HNET+R experiments reported on the main text we use VAEs as our replay models (Fig. A1a, Kingma & Welling, 2014). Briefly, a VAE consists of an encoder-decoder network pair, where the encoder network processes some input pattern $\mathbf{x}$ and its outputs $f_{\mathrm{enc}}(\mathbf{x}) = (\boldsymbol{\mu}, \boldsymbol{\sigma}^2)$ comprise the parameters $\boldsymbol{\mu}$ and $\boldsymbol{\sigma}^2$ (encoded in $\log$ domain, to enforce nonnegativity) of a diagonal multivariate Gaussian $p_Z(\mathbf{z}; \boldsymbol{\mu}, \boldsymbol{\sigma}^2)$, which governs the distribution of latent samples $\mathbf{z}$. On the other side of the circuit, the decoder network processes a latent sample $\mathbf{z}$ and a one-hot-encoded task identity vector and returns an input pattern reconstruction, $f_{\mathrm{dec}}(\mathbf{z}, \mathbf{1}_t) = \hat{\mathbf{x}}$.

VAEs can preserve memories using a technique called generative replay: when training task $T$, input samples are generated from the current replay network for old tasks $t < T$, by varying $\mathbf{1}_t$ and drawing latent space samples $\mathbf{z}$. Generated data can be mixed with the current dataset, yielding an augmented dataset $\tilde{\mathcal{X}}$ used to relearn model parameters. When protecting a discriminative model, synthetic 'soft' targets can be generated by evaluating the network on $\tilde{\mathcal{X}}$. We use this strategy to protect an auxiliary task inference classifier in HNET+TIR, and to protect the main target model in HNET+R.

**Hypernetwork-protected replay.** In our HNET+TIR and HNET+R experiments, we parameterize the decoder network through a task-conditioned hypernetwork, $f_{\mathrm{h,dec}}(\mathbf{e}, \Theta_{\mathrm{h,dec}})$. In combination with our output regularizer, this allows us to take advantage of the memory retention capacity of hypernetworks, now on a generative model.

The replay model (encoder, decoder and decoder hypernetwork) is a separate subsystem that is optimized independently from the target network. Its parameters $\Theta_{\mathrm{enc}}$ and $\Theta_{\mathrm{h,dec}}$ are learned by minimizing our regularized loss function, Eq. 2, here with the task-specific term set to the standard VAE objective function,

$$\mathcal{L}_{\mathrm{VAE}}(\mathbf{X}, \Theta_{\mathrm{enc}}, \Theta_{\mathrm{h,dec}}) = \mathcal{L}_{\mathrm{rec}}(\mathbf{X}, \Theta_{\mathrm{enc}}, \Theta_{\mathrm{dec}}) + \mathcal{L}_{\mathrm{prior}}(\mathbf{X}, \Theta_{\mathrm{enc}}, \Theta_{\mathrm{dec}}), \tag{3}$$

with $\Theta_{\mathrm{dec}} = f_{\mathrm{h,dec}}(\mathbf{e}, \Theta_{\mathrm{h,dec}})$ introducing the dependence on $\Theta_{\mathrm{h,dec}}$. $\mathcal{L}_{\mathrm{VAE}}$ balances a reconstruction $\mathcal{L}_{\mathrm{rec}}$ and a prior-matching $\mathcal{L}_{\mathrm{prior}}$ penalties. For our MNIST experiments, we choose binary cross-entropy (in pixel space) as the reconstruction loss, that we write below for a single example $\mathbf{x}$

$$\mathcal{L}_{\mathrm{rec}}(\mathbf{x}, \Theta_{\mathrm{enc}}, \Theta_{\mathrm{dec}}) = \mathcal{L}_{\mathrm{xent}}\left(\mathbf{x}, f_{\mathrm{dec}}\left(\mathbf{z}, \mathbf{1}_{t(\mathbf{x})}, \Theta_{\mathrm{dec}}\right)\right), \tag{4}$$

where $\mathcal{L}_{\mathrm{xent}}(t, y) = -\sum_k t_k \log y_k$ is the cross entropy. For a diagonal Gaussian $p_Z$, the prior-matching term can be evaluated analytically,

$$\mathcal{L}_{\mathrm{prior}} = -\frac{1}{2} \sum_{i=1}^{|\mathbf{z}|} \left(1 + \log \sigma_i^2 - \sigma_i^2 - \mu_i^2\right). \tag{5}$$

Above, $\mathbf{z}$ is a sample from $p_Z(\mathbf{z}; \boldsymbol{\mu}(\tilde{\mathbf{x}}), \boldsymbol{\sigma}^2(\tilde{\mathbf{x}}))$ obtained via the reparameterization trick (Kingma & Welling, 2014; Rezende et al., 2014). This introduces the dependency of $\mathcal{L}_{\mathrm{rec}}$ on $\Theta_{\mathrm{enc}}$.

**Task inference network (HNET+TIR).** In the HNET+TIR setup, we extend our system to include a task inference neural network classifier $\boldsymbol{\alpha}(\mathbf{x})$ parameterized by $\Theta_{\mathrm{TI}}$, where tasks are encoded with a $T$-dimensional softmax output layer. In both CL2 and CL3 scenarios we use a growing single-head setup for $\boldsymbol{\alpha}$, and increase the dimensionality of the softmax layer as tasks arrive.

This network is prone to catastrophic forgetting when tasks are learned continually. To prevent this from happening we resort to replay data generated from a hypernetwork-protected VAE, described above. More specifically, we introduce a task inference loss,

$$\mathcal{L}_{\text{TI}}(\tilde{\mathbf{x}}, \Theta_{\text{TI}}) = \mathcal{L}_{\text{xent}}(\mathbf{1}_{t(\tilde{\mathbf{x}})}, \boldsymbol{\alpha}(\tilde{\mathbf{x}}, \Theta_{\text{enc}})), \tag{6}$$

where $t(\tilde{\mathbf{x}})$ denotes the correct task identity for a sample $\tilde{\mathbf{x}}$ from the augmented dataset $\tilde{\mathcal{X}} = \{\tilde{\mathbf{X}}^{(1)}, \dots \tilde{\mathbf{X}}^{(T-1)}, \tilde{\mathbf{X}}^{(T)}\}$ with $\tilde{\mathbf{X}}^{(t)}$ being synthetic data $f_{\text{dec}}(\mathbf{z}, \mathbf{1}_t, \Theta_{\text{dec}})$ for $t = 1 \dots T-1$ and $\tilde{\mathbf{X}}^{(T)} = \mathbf{X}^{(T)}$ is the current task data. Importantly, synthetic data is essential to obtain a well defined objective function for task inference; the cross-entropy loss $\mathcal{L}_{\text{TI}}$ requires at least two groundtruth classes to be optimized. Note that replayed data can be generated online by drawing samples $z$ from the prior.

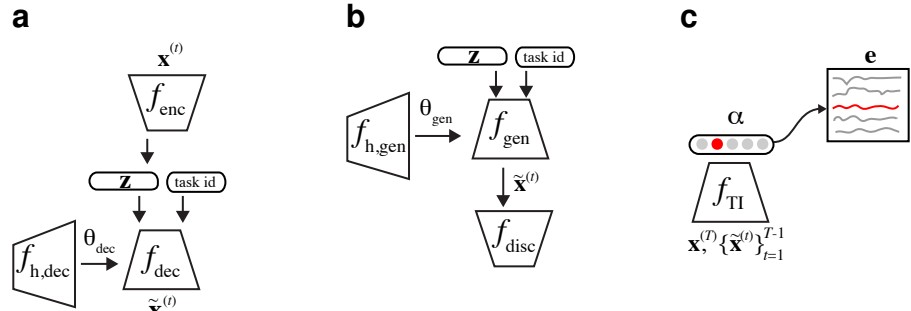

Figure A1: **Hypernetwork-protected replay model setups.** **(a)** A hypernetwork-protected VAE, that we used for HNET+R and HNET+TIR main text experiments. **(b)** A hypernetwork-protected GAN, that we used for our class-incremental learning Appendix F experiments. **(c)** A task inference classifier protected with synthetic replay data, used on HNET+TIR experiments.

**Hypernetwork-protected GANs.**   Generative adversarial networks (Goodfellow et al., 2014) have become an established method for generative modelling and tend to produce higher quality images compared to VAEs, even at the scale of datasets as complex as ImageNet (Brock et al., 2019; Lučić et al., 2019; Donahue & Simonyan, 2019). This makes GANs perfect candidates for powerful replay models. A suitable GAN instantiation for CL is the conditional GAN (Mirza & Osindero, 2014) as studied by Wu et al. (2018). Recent developments in the GAN literature already allude towards the potential of using hypernetwork-like structures, e.g., when injecting the latent noise (Karras et al., 2019) or when using class-conditional batch-normalization as in (Brock et al., 2019). We propose to go one step further and use a hypernetwork that maps the condition to the full set of generator parameters $\Theta_{\text{gen}}^*$. Our framework allows training a conditional GAN one condition at the time. This is potentially of general interest, and goes beyond the scope of replay models, since conditional GANs trained in a mutli-task fashion as in Brock et al. (2019) require very large computational resources.

For our showcase experiment on class-incremental MNIST learning, Fig. A8, we did not aim to compare to related work and therefore did not tune to have less weights in the hypernetwork than on the target network (for the VAE experiments, we use the same compressive setup as in the main text, see Appendix C). The GAN hypernetwork is a fully-connected chunked hypernetwork with 2 hidden layers of size 25 and 25 followed by an output size of 75,000. We used learning rates for both discriminator and the generator hypernetwork of 0.0001, as well as dropout of 0.4 in the discriminator and the system is trained for 10000 iterations per task. We use the Pearson Chi$^2$ Least-Squares GAN loss from Mao et al. (2017) in our experiments.

## C   ADDITIONAL EXPERIMENTAL DETAILS

All experiments are conducted using 16 NVIDIA GeForce RTX 2080 TI graphics cards.

For simplicity, we decided to always keep the previous task embeddings $\mathbf{e}^{(t)}$, $t = 1, \dots, T-1$, fixed and only learn the current task embedding $\mathbf{e}^{(T)}$. In general, performance should be improved if the

regularizer in Eq. 2 has a separate copy of the task embeddings $\mathbf{e}^{(t,*)}$ from before learning the current task, such that $\mathbf{e}^{(t)}$ can be adapted. Hence, the targets become $f_\mathrm{h}(\mathbf{e}^{(t,*)}, \Theta_\mathrm{h}^*)$ and remain constant while learning task $T$. This would give the hypernetwork the flexibility to adjust the embeddings i.e. the preimage of the targets and therefore represent any function that includes all desired targets in its image.

**Nonlinear regression toy problem.** The nonlinear toy regression from Fig. 2 is an illustrative example for a continual learning problem where a set of ground-truth functions $\{g^{(1)}, \ldots, g^{(T)}\}$ is given from which we collect 100 noisy training samples per task $\{(\mathbf{x}, \mathbf{y}) \mid \mathbf{y} = g^{(t)}(\mathbf{x}) + \epsilon \text{ with } \epsilon \sim \mathcal{N}(0, \sigma^2 I), \mathbf{x} \sim \mathcal{U}(\mathcal{X}^{(t)})\}$, where $\mathcal{X}^{(t)}$ denotes the input domain of task $t$. We set $\sigma = 0.05$ in this experiment.

We perform 1D regression and choose the following set of tasks:

$$g^{(1)}(x) = x + 3 \qquad\qquad \mathcal{X}^{(1)} = [-4, -2] \tag{7}$$

$$g^{(2)}(x) = 2x^2 - 1 \qquad\qquad \mathcal{X}^{(2)} = [-1, 1] \tag{8}$$

$$g^{(3)}(x) = (x - 3)^3 \qquad\qquad \mathcal{X}^{(3)} = [2, 4] \tag{9}$$

The target network $f_\mathrm{trgt}$ consists of two fully-connected hidden layers using 10 neurons each. For illustrative purposes we use a full hypernetwork $f_\mathrm{h}$ that generates all 141 weights of $f_\mathrm{trgt}$ at once, also being a fully-connected network with two hidden-layers of size 10. Hence, this is the only setup where we did not explore the possibility of a chunked hypernetwork. We use sigmoid activation functions in both networks. The task embedding dimension was set to 2.

We train each task for 4000 iterations using the Adam optimizer with a learning rate of 0.01 (and otherwise default PyTorch options) and a batch size of 32.

To test our regularizer in Fig. 2a we set $\beta_\mathrm{output}$ to 0.005, while it is set to 0 for the fine-tuning experiment in Fig. 2c.

For the multi-task learner in Fig. 2b we trained only the target network (no hypernetwork) for 12000 iterations with a learning rate of 0.05. Comparable performance could be obtained when training the task-conditioned hypernetwork in this multi-task regime (data not shown).

It is worth noting that the multi-task learner from Fig. 2b that uses no hypernetwork is only able to learn the task since we choose the input domains to be non-overlapping.

**Permuted MNIST benchmark.** For our experiments conducted on MNIST we replicated the experimental setup proposed by van de Ven & Tolias (2019) whenever applicable. We therefore use the same number of training iterations, the same or a lower number of weights in the hypernetwork than in the target network, the same learning rates and the same optimizer. For the replay model, i.e., the hypernetwork-empowered VAE, as well as for the standard classifier we used 5000 training iterations per task and learning rate is set to 0.0001 for the Adam optimizer (otherwise PyTorch default values). The batchsize is set to 128 for the VAE whereas the classifier is simultaneously trained on a batch of 128 samples of replayed data (evenly distributed over all past tasks) and a batch of 128 images from the currently available dataset. MNIST images are padded with zeros, which results in network inputs of size $32 \times 32$, again strictly following the implementation of the compared work. We experienced better performance when we condition our replay model on a specific task input. We therefore construct for every task a specific input namely a sample from a standard multivariate normal of dimension 100. In practice we found the dimension to be not important. This input stays constant throughout the experiment and is not learned. Note that we use the same hyperparameters for all learning scenarios, which is not true for the reported related work since they have tuned special hyperparameters for all scenarios and all methods.

- **Details of hypernetwork for the VAE.** We use one hypernetwork configuration to generate weights for all variational autoencoders used for our PermutedMNIST-10 experiments namely a fully-connected chunked hypernetwork with 2 hidden layers of size 25 and 25 followed by an output size of 85,000. We use ELU nonlinearities in the hidden layers

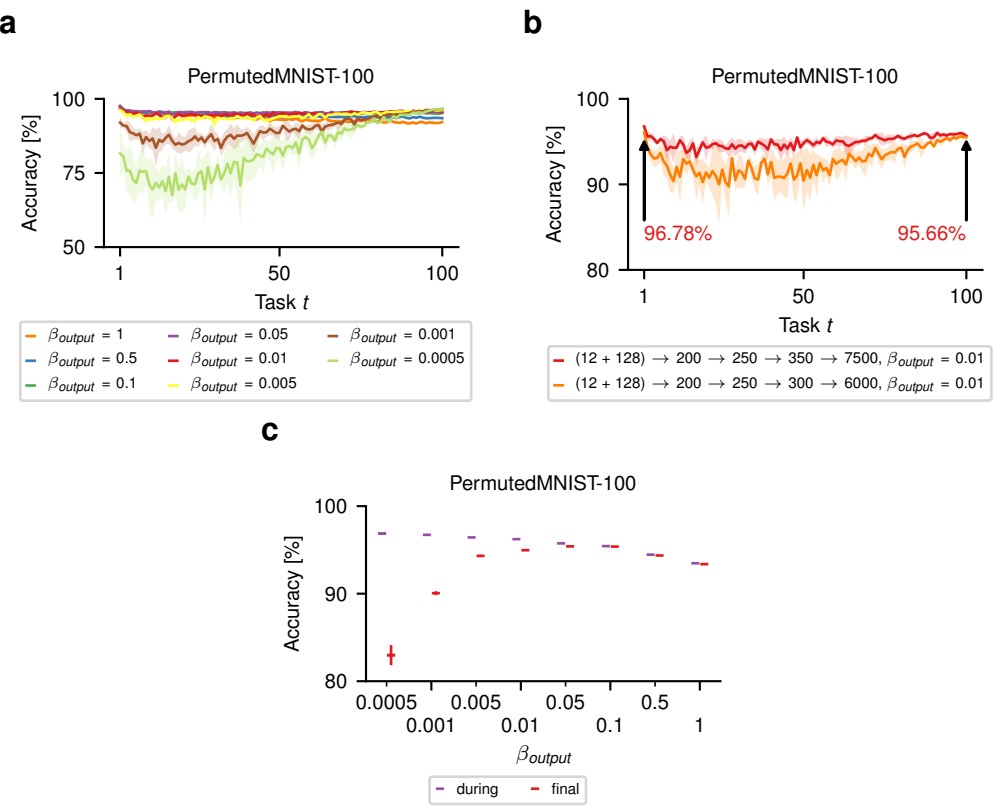

Figure A2: **Additional experiments on the PermutedMNIST-100 benchmark. (a)** Final test set classification accuracy on the $t$-th task after learning one hundred permutations (PermutedMNIST-100). All runs use exactly the same hyperparameter configuration except for varying values of $\beta_{\text{output}}$. The final accuracies are robust for a wide range of regularization strengths. If $\beta_{\text{output}}$ is too weak, forgetting will occur. However, there is no severe disadvantage of choosing $\beta_{\text{output}}$ too high (cmp. (c)). A too high $\beta_{\text{output}}$ simply shifts the attention of the optimizer away from the current task, leading to lower baseline accuracies when the training time is not increased. **(b)** Due to an increased number of output neurons, the target network for PermutedMNIST-100 has more weights than for PermutedMNIST-10 (this is only the case for CL1 and CL3). This plot shows that the performance drop is minor when choosing a hypernetwork with a comparable number of weights as the target network in CL2 (orange) compared to one that has a similar number of weights as the target network for CL1 in PermutedMNIST-100 (red). **(c)** Task-averaged test set accuracy after learning all tasks (labelled 'final', in red) and immediately after learning a task (labelled 'during', in purple) for the runs depicted in (a). For low values of $\beta_{\text{output}}$ final accuracies are worse than immediate once (forgetting occurs). If $\beta_{\text{output}}$ is too high, baseline accuracies decrease since the optimizer puts less emphasis on the current task (note that the training time per task is not increased). Shaded areas in (a) and (b) denote STD, whereas error bars in (c) denote SEM (always across 5 random seeds).

of the hypernetwork. The size of task embeddings $\mathbf{e}$ has been set to 24 and the size of chunk embeddings $\mathbf{c}$ to 8. The parameter $\beta_{\text{output}}$ is 0.05 . The number of weights in this hypernetwork is 2,211,907 (2,211,691 network weights + 216 task embedding weights). The corresponding target network (and therefore output of the chunked hypernetwork), as taken from related work, has 2,227,024 weights.

- **Details of the VAE for HNET+TIR.** For this variational autoencoder, we use two fully-connected neural networks with layers of size 1000, 1000 for the encoder and 1000, 1000 for the decoder and a latent space of 100. This setup is again copied from work we compare against.

- **Details of the VAE for HNET+R.** For this variational autoencoder, we use two fully-connected neural networks with layers of size 400, 400 for the encoder and 400, 400 for the decoder (both 1000, 1000 in the related work) and a latent space of dimension 100. Here, we departure from related work by choosing a smaller architecture for the autoencoder. Note that we still use a hypernetwork with less trainable parameters than the target network (in this case the decoder) that is used in related work.

- **Details of the hypernetwork for the target classifier in PermutedMNIST-10 (HNET+TIR & HNET+ENT).** We use the same setup for the hypernetwork as used for the VAEs above, but since the target network is smaller we reduce the output of the hypernetwork to 78,000. We also adjust the parameter $\beta_{\text{output}}$ to 0.01, consistent with our PermutedMNIST-100 experiments. The number of weights in this hypernetwork is therefore 2,029,931 parameters (2,029,691 network weights + 240 task embedding weights). The corresponding target network (from related work) would have 2,126,100 weights for `CL1` and `CL3` and 2,036,010 for `CL2` (only one output head).

- **Details of the hypernetwork for the target classifier for PermutedMNIST-100.** For these experiments we chose an architecture that worked well on the PermutedMNIST-10 benchmark and did not conduct any more search for new architectures. For PermutedMNIST-100, the reported results were obtained by using a chunked hypernetwork with 3 hidden layers of size 200, 250 and 350 (300 for `CL2`) and an output size of 7500 (6000 for `CL2`) (such that we approximately match the corresponding target network size for `CL1`/`CL2`/`CL3`). Interestingly, Fig. A2b shows that even if we don't adjust the number of hypernetwork weights to the increased number of target network weights, the superiority of our method is evident. Aside from this, the plots in Fig. 3 have been generated using the PermutedMNIST-10 HNET+TIR setup (note that this includes the conditions set by related work for PermutedMNIST-10, e.g., target network sizes, the number of training iterations, learning rates, etc.).

- **Details of the VAE and the hypernetwork for the VAE in PermutedMNIST-100 for** `CL2`/`CL3`. We use a very similar setup for the VAE and it's hypernetwork used in HNET+TIR for PermutedMNIST-10 as described above. We only applied the following changes: Fully-connected hypernetwork with one hidden layer of size 100; chunk embedding sizes are set to 12; task embedding sizes are set two 128 and the hidden layer sizes of the VAE its generator are 400, 600. Also we increased the regularisation strength $\beta_{\text{output}} = 0.1$ for the VAE its generator hypernetwork.

- **Details of the target classifier for HNET+TIR & HNET+ENT.** For this classifier, we use the same setup as in the study we compare to (van de Ven & Tolias, 2019), i.e., a fully-connected network with layers of size 1000, 1000. Note that if the classifier is used as a task inference model, it is trained on replay data and the corresponding hard targets, i.e., the argmax of the soft targets.

Below, we report the specifications for our automatic hyperparameter search (if not noted otherwise, these specifications apply for the split MNIST and split CIFAR experiments as well):

- Hidden layer sizes of the hypernetwork: (no hidden layer), "5,5" "10,10", "25,25", "50,50", "100,100", "10", "50", "100"
- Output size of the hypernetwork: fitted such that we obtain less parameters then the target network which we compare against
- Embedding sizes (for $\mathbf{e}$ and $\mathbf{c}$): 8, 12, 24, 36, 62, 96, 128
- $\beta_{\text{output}}$: 0.0005, 0.001, 0.005, 0.01, 0.005, 0.1, 0.5, 1.0

- Hypernetwork transfer functions: linear, ReLU, ELU, Leaky-ReLU

Note that only a random subset of all possible combinations of hyperparameters has been explored.

After we found a configuration with promising accuracies and a similar number of weights compared to the original target network, we manually fine-tuned the architecture to increase/decrease the number of hypernetwork weights to approximately match the number of target network weights.

The choice of hypernetwork architecture seems to have a strong influence on the performance. It might be worth exploring alternatives, e.g., an architecture inspired by those used in typical generative models. We note that in addition to the above specifications we explored manually some hyperparameter configurations to gain a better understanding of our method.

**Split MNIST benchmark.** Again, whenever applicable we reproduce the setup from van de Ven & Tolias (2019). Differences to the PermutedMNIST-10 experiments are just the learning rate (0.001) and the number of training iterations (set to 2000).

- **Details of hypernetwork for the VAE.** We use one hypernetwork configuration to generate weights for all variational autoencoders used for our split MNIST experiments, namely a fully-connected chunked hypernetwork with 2 hidden layers of size 10, 10 followed by an output size of 50,000. We use ELU nonlinearities in the hidden layers of the hypernetwork. The size of task embeddings $\mathbf{e}$ has been set to 96 and the size of chunk embeddings $\mathbf{c}$ to 96. The parameter $\beta_{\text{output}}$ is 0.01 for HNET+R and 0.05 for HNET+TIR . The number of weights in this hypernetwork is 553,576 (553,192 network weights + 384 task embedding weights). The corresponding target network (and therefore output of the chunked hypernetwork), as taken from related work, has 555,184 weights. For a qualitative analyses of the replay data of this VAE (class incrementally learned), see A8.

- **Details of the VAE for HNET+TIR.** For this variational autoencoder, we use two fully-connected neural networks with layers of size 400, 400 for the encoder and 50, 150 for the decoder (both 400, 400 in the related work) and a latent space of dimension 100.

- **Details of the VAE for HNET+R.** For this variational autoencoder, we use two fully-connected neural networks with layers of size 400, 400 for the encoder and 250, 350 for the decoder (both 400, 400 in the related work) and a latent space of dimension 100.

- **Details of the hypernetwork for the target classifier in split MNIST (HNET+TIR & HNET+ENT).** We use the same setup for the hypernetwork as used for the VAE above, but since the target network is smaller we reduce the output of the hypernetwork to 42,000. We also adjust the $\beta_{\text{output}}$ to 0.01 although this parameter seems to not have a strong effect on the performance. The number of weights in this hypernetwork is therefore 465,672 parameters (465,192 network weights + 480 task embedding weights). The corresponding target network (from related work) would have 478,410 weights for `CL1` and `CL3` and 475,202 for `CL2` (only one output head).

- **Details of the target classifier for HNET+TIR & HNET+ENT.** For this classifier, we again use the same setup as in the study we compare to (van de Ven & Tolias, 2019), i.e., a fully-connected neural networks with layers of size 400, 400. Note that if the classifier is used as a task inference model, it is trained on replay data and the corresponding hard targets, i.e., the argmax the soft targets.

**Split CIFAR-10/100 benchmark.** For these experiments, we used as a target network a ResNet-32 network (He et al. (2016)) and again produce the weights of this target network by a hypernetwork in a compressive manner. The hypernetwork in this experiment directly maps from the joint task and chunk embedding space (both dimension 32) to the output space of the hypernetwork, which is of dimension 7,000. This hypernetwork has 457,336 parameters (457,144 network weights + 192 task embedding weights). The corresponding target network, the ResNet-32, has 468.540 weights (including batch-norm weights). We train for 200 epochs per task using the Adam optimizer with an initial learning rate of 0.001 (and otherwise default PyTorch values) and a batch size of 32. In addition, we apply the two learning rate schedules suggested in the Keras CIFAR-10 example[2].

---

[2]See https://keras.io/examples/cifar10_resnet/.

Due to the use of batch normalization, we have to find an appropriate way to handle the running statistics which are estimated during training. Note, these are not parameters which are trained through backpropagation. There are different ways how the running statistics could be treated:

1. One could ignore the running statistics altogether and simply compute statistics based on the current batch during evaluation.

2. The statistics could be part of the hypernetwork output. Therefore, one would have to manipulate the target hypernetwork output of the previous task, such that the estimated running statistics of the previous task will be distilled into the hypernetwork.

3. The running statistics can simply be checkpointed and stored after every task. Note, this method would lead to a linear memory growth in the number of tasks that scales with the number of units in the target network.

For simplicity, we chose the last option and simply checkpointed the running statistics after every task.

For the fine-tuning results in Fig. 5 we just continually updated the running statistics (thus, we applied no checkpointing).

## D  ADDITIONAL EXPERIMENTS AND NOTES

**Split CIFAR-10/100 benchmark using the model of Zenke et al. (2017).**  We re-run the split CIFAR-10/100 experiment reported on the main text while reproducing the setup from Zenke et al. (2017). Our overall classification performance is comparable to synaptic intelligence, which achieves 73.85% task-averaged test set accuracy, while our method reaches 71.29% $\pm$ 0.32%, with initial baseline performance being slightly worse in our approach, Fig. A3.

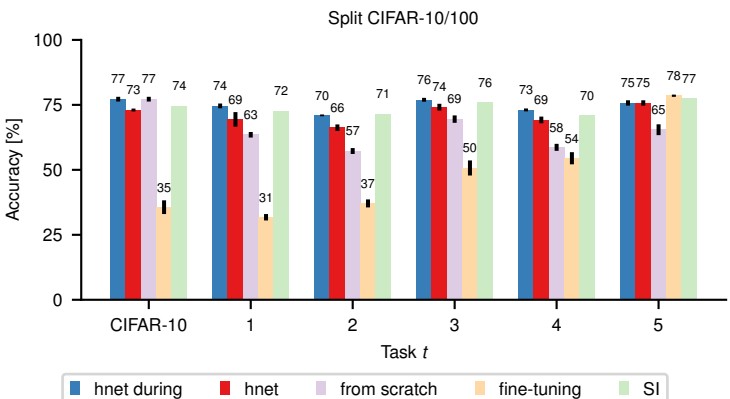

Figure A3: **Replication of the split CIFAR-10/100 experiment of Zenke et al. (2017).** Test set accuracies on the entire CIFAR-10 dataset and subsequent CIFAR-100 splits. Both task-conditioned hypernetworks (hnet, in red) and synaptic intelligence (SI, in green) transfer information forward and are protected from catastrophic forgetting. The performance of the two methods is comparable. For completeness, we report our test set accuracies achieved immediately after training (hnet-during, in blue), when training from scratch (purple), and with our regularizer turned off (fine-tuning, yellow).

To obtain our results, we use a hypernetwork with 3 hidden-layers of sizes 100, 150, 200 and output size 5500. The size of task embeddings **e** has been set to 48 and the size of chunk embeddings **c** to 80. The parameter $\beta_{\text{output}}$ is 0.01 and the learning rate is set to 0.0001.

The number of weights in this hypernetwork is 1,182,678 (1,182,390 network weights + 288 task embedding weights). The corresponding target network would have 1,276,508 weights.

In addition to the above specified hyperparameter search configuration we also included the following learning rates: 0.0001, 0.0005, 0.001 and manually tuned some architectural parameters.

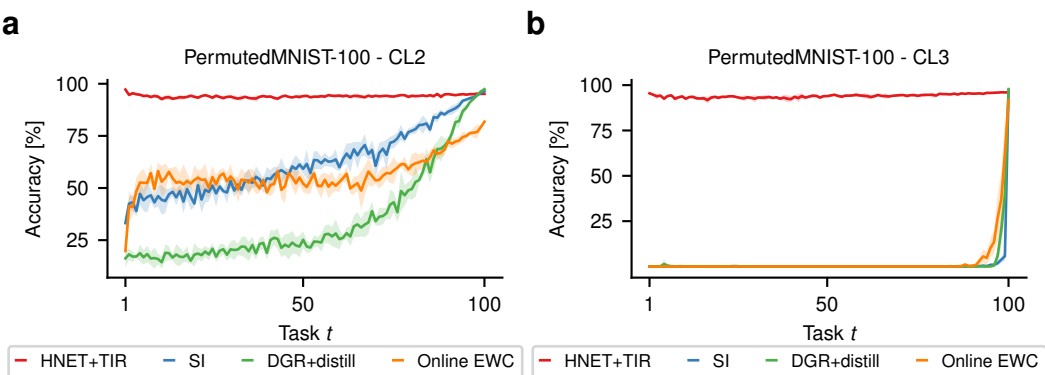

Figure A4: **Context-free inference using hypernetwork-protected replay (HNET+TIR) on long task sequences.** Final test set classification accuracy on the $t$-th task after learning one hundred permutations of the MNIST dataset (PermutedMNIST-100) for the CL2 (a) and CL3 (b) scenarios, where task identity is not explicitly provided to the system. As before, the number of hypernetwork parameters is not larger than that of the related work we compare to. (**a**) HNET+TIR displays almost perfect memory retention. We used a stochastic regularizer (cf. Appendix D note below) which evaluates the output regularizer in Eq. 2 only for a random subset of previous tasks (here, twenty). (**b**) HNET+TIR is the only method that is capable of learning PermutedMNIST-100 in this learning scenario. For this benchmark, the input data domains are easily separable and the task inference system achieves virtually perfect (~100%) task inference accuracy throughout, even for this long experiment. HNET+TIR uses a divide-and-conquer strategy: if task inference is done right, CL3 becomes just CL1. Furthermore, once task identity is predicted, the final softmax computation only needs to consider the corresponding task outputs in isolation (here, of size 10). Curiously, for HNET+TIR, CL2 can be harder than CL3 as the single output layer (of size 10, shared by all tasks) introduces a capacity bottleneck. The related methods, on the other hand, have to consider the entire output layer (here, of size 10*100) at once, which is known to be harder to train sequentially. This leads to overwhelming error rates on long problems such as PermutedMNIST-100. Shaded areas in (a) and (b) denote STD ($n = 5$).

**Upper bound for replay models.** We obtain an upper bound for the replay-based experiments (Table 2) by sequentially training a classifier, in the same way as for HNET+R and DGR, now using true input data from past tasks and a synthetic, self-generated target. This corresponds to the rehearsal thought experiment delineated in Sect. 1.

Table 2: Task-averaged test accuracy ($\pm$ SEM, $n = 20$) on the permuted ('P10') and split ('S') MNIST experiments. For HNET+R and DGR+distill (van de Ven & Tolias, 2019) the classification network is trained sequentially on data from the current task and replayed data from all previous tasks. Our HNET+R comes close to saturating the corresponding replay upper bound RPL-UB.

|         | DGR              | HNET+R           | RPL-UB           |
|---------|------------------|------------------|------------------|
| P10-CL1 | $97.51 \pm 0.01$ | $97.85 \pm 0.02$ | $97.89 \pm 0.02$ |
| P10-CL2 | $97.35 \pm 0.02$ | $97.60 \pm 0.02$ | $97.72 \pm 0.01$ |
| P10-CL3 | $96.38 \pm 0.03$ | $97.71 \pm 0.06$ | $97.91 \pm 0.01$ |
| S-CL1   | $99.61 \pm 0.02$ | $99.81 \pm 0.01$ | $99.83 \pm 0.01$ |
| S-CL2   | $96.83 \pm 0.20$ | $97.88 \pm 0.05$ | $98.96 \pm 0.03$ |
| S-CL3   | $91.79 \pm 0.32$ | $94.97 \pm 0.18$ | $98.38 \pm 0.02$ |

**Quantification of forgetting in our continual learning experiments.** In order to quantify forgetting of our approach, we compare test set accuracies of every single task directly after training with it's test set accuracy after training on all tasks.

Only CL1 is shown since other scenarios i.e. CL2 and CL3 depend on task inference which only is measurable after training on all tasks.

Table 3: Task-averaged test accuracy ($\pm$ SEM, $n = 20$) on the permutedMNIST-10 ('P10') and splitMNIST ('S') experiments during and after training.

|         | HNET+TIR during  | HNET+TIR after   | HNET+R during    | HNET+R after     |
|---------|------------------|------------------|------------------|------------------|
| S-CL1   | $99.79 \pm 0.01$ | $99.79 \pm 0.01$ | $99.82 \pm 0.01$ | $99.83 \pm 0.01$ |
| P10-CL1 | $97.58 \pm 0.02$ | $97.57 \pm 0.02$ | $98.03 \pm 0.01$ | $97.87 \pm 0.01$ |

Table 4: Task-averaged test accuracy ($\pm$ SEM, $n = 5$) on the permutedMNIST-100 ('P100') experiments during and after training.

|          | HNET+TIR during  | HNET+TIR after   |
|----------|------------------|------------------|
| P100-CL1 | $96.12 \pm 0.08$ | $96.18 \pm 0.09$ |
| P100-CL2 | -                | $95.97 \pm 0.05$ |
| P100-CL3 | -                | $96.00 \pm 0.03$ |

Table 5: Task-averaged test accuracy ($\pm$ SEM, $n = 5$) on split CIFAR-10/100 on CL1 on two different target network architectures.

|           | during           | after            |
|-----------|------------------|------------------|
| ZenkeNet  | $74.75 \pm 0.09$ | $71.29 \pm 0.32$ |
| ResNet-32 | $82.36 \pm 0.44$ | $82.34 \pm 0.44$ |

**Robustness of $\beta_{\text{output}}$-choice.** In Fig. A2a and Fig. A2c we provide additional experiments for our method on PermutedMNIST-100. We show that our method performs comparable for a wide range of $\beta_{\text{output}}$-values (including the one depicted in Fig. 3a).

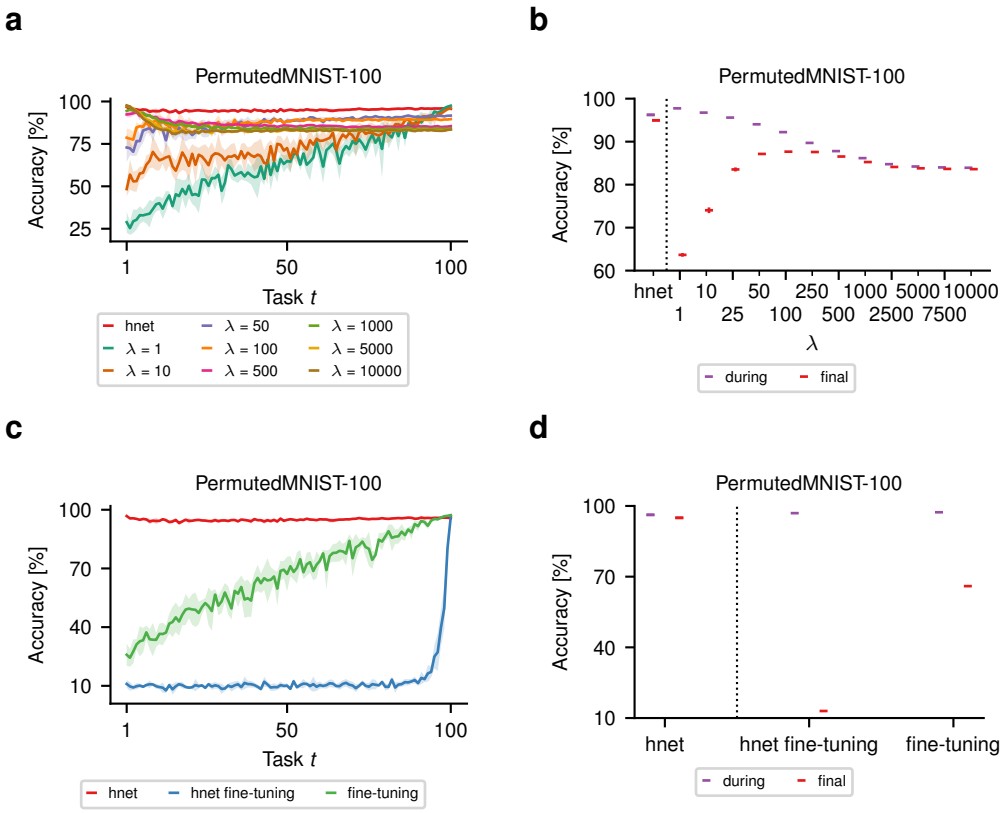

Figure A5: **Additional experiments with online EWC and fine-tuning on the PermutedMNIST-100 benchmark. (a)** Final test set classification accuracy on the $t$-th task after learning one hundred permutations (PermutedMNIST-100) using the online EWC algorithm (Schwarz et al., 2018) to prevent forgetting. All runs use exactly the same hyperparameter configuration except for varying values of the regularization strength $\lambda$. Our method (hnet, in red) and the online EWC run ($\lambda = 100$, in orange) from Fig. 3a are shown for comparison. It can be seen that even when tuning the regularization strength one cannot attain similar performance as with our approach (cmp. Fig. A2a). Too strong regularization prevents the learning of new tasks whereas too weak regularization doesn't prevent forgetting. However, a middle ground (e.g., using $\lambda = 100$) does not reach acceptable per-task performances. **(b)** Task-averaged test set accuracy after learning all tasks (labelled 'final', in red) and immediately after learning a task (labelled 'during', in purple) for a range of regularization strengths $\lambda$ when using the online EWC algorithm. Results are complementary to those shown in (a). **(c)** Final test set classification accuracy on the $t$-th task after learning one hundred permutations (PermutedMNIST-100) when applying fine-tuning to the hypernetwork (labelled 'hnet fine-tuning', in blue) or target network (labelled 'fine-tuning', in green). Our method (hnet, in red) from Fig. 3a is shown for comparison. It can be seen that without protection the hypernetwork suffers much more severely from catastrophic forgetting as when training a target network only. **(d)** This plot is complementary to (c). See description of (b) for an explanation of the labels. Shaded areas in (a) and (c) denote STD, whereas error bars in (b) and (d) denote SEM (always across 5 random seeds).

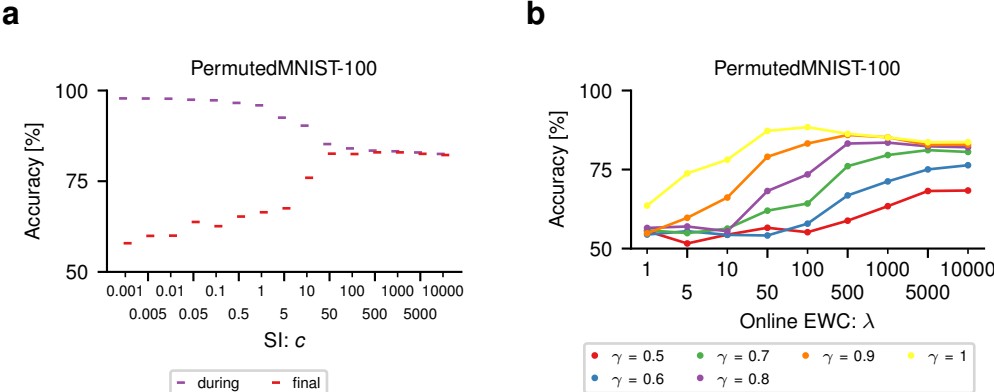

Figure A6: **Hyperparameter search for online EWC and SI on the PermutedMNIST-100 benchmark.** We conduct the same hyperparameter search as performed in van de Ven & Tolias (2018). We did not compute different random seeds for this search. **(a)** Hyperparameter search on the regularisation strength $c$ for the SI algorithm. Accuracies during and after the experiment are shown. **(b)** Hyperparameter search for parameters $\lambda$ and $\gamma$ of the online EWC algorithm. Only accuracies after the experiment are shown.

**Varying the regularization strength for online EWC.** The performance of online EWC in Fig. 3a is closest to our method (labelled hnet, in red) compared to the other methods. Therefore, we take a closer look at this method and show that further adjustments of the regularization strength $\lambda$ do not lead to better performance. Results for a wide range of regularization strengths can be seen in Fig. A5a and Fig. A5b. As shown, online EWC cannot attain a performance comparable to our method when tuning the regularization strength only.

**The impact of catastrophic forgetting on the hypernetwork and target network.** We have successfully shown that by shifting the continual learning problem from the target network to the hypernetwork we can successfully overcome forgetting due to the introduction of our regularizer in Eq. 2. We motivated this success by claiming that it is an inherently simpler task to remember a few input-output mappings in the hypernetwork (namely the weight realizations of each task) rather than the massive number of input-output mappings $\{(\mathbf{x}^{(t,i)}, \mathbf{y}^{(t,i)})\}_{i=1}^{n_t}$ associated with the remembering of each task $t$ by the target network.

Further evidence of this claim is provided by fine-tuning experiments in Fig. A5c and Fig. A5d. Fine-tuning refers to sequentially learning a neural network on a set of tasks without any mechanism in place to prevent forgetting. It is shown that fine-tuning a target network (no hypernetwork in this setup) has no catastrophic influence on the performance of previous tasks. Instead there is a graceful decline in performance. On the contrary, catastrophic forgetting has an almost immediate affect when training a hypernetwork without protection (i.e., training our method with $\beta_{\text{output}} = 0$. The performance quickly drops to chance level, suggesting that if we weren't solving a simpler task then preventing forgetting in the hypernetwork rather than in the target network might not be beneficial.

**Chunking and hypernetwork architecture sensitivity.** In this note we investigate the performance sensitivity for different (fully-connected) hypernetwork architectures on split MNIST and PermutedMNIST-10, Fig. A7. We trained thousands of randomly drawn architectures from the following grid (the same training hyperparameters as reported for for CL1, see Appendix C, were used throughout): possible number of hidden layers $1, 2$, possible layer size $5, 10, 20, \ldots, 90, 100$, possible chunk embedding size $8, 12, 24, 56, 96$ and hypernetwork output size in $\{10, 50, 100, 200, 300, 400, 500, 750, 1k, 2k, \ldots, 9k, 10k, 20k, 30k, 40k\}$. Since we realize compression through chunking, we sort our hypernetwork architectures by compression ratio, and consider only architectures with small compression ratios.

Performance of split MNIST stays in the high 90 percentages even when reaching compression ratios close to $1\%$ whereas for PermutedMNIST-10 accuracies decline in a non-linear fashion. For both experiments, the choice of the chunked hypernetwork archicture is robust and high performing even in

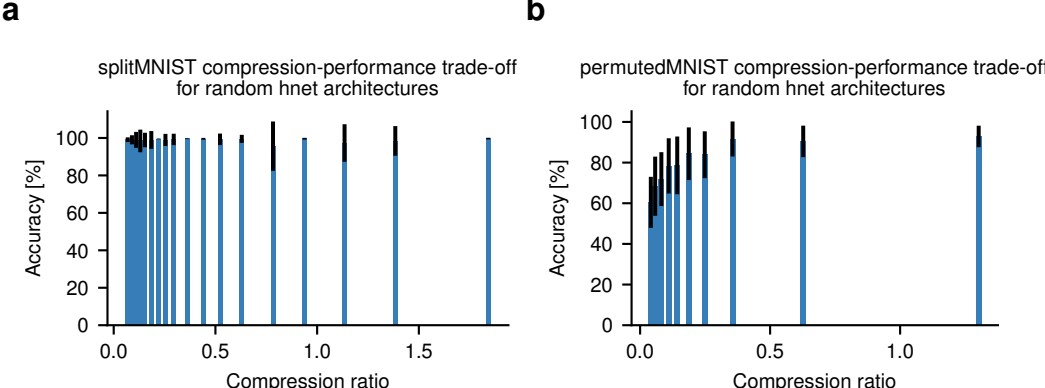

Figure A7: **Robustness to hypernetwork architecture choice for a large range of compression ratios.** Performance vs. compression for random hypernetwork architecture choices, for split MNIST and PermutedMNIST-10 (mean $\pm$ STD, $n = 500$ architectures per bin). Every model was trained with the same setup (including all hyperparameters) used to obtain results reported in Table 1 (CL1). We considered architectures yielding compression ratios $|\Theta_\text{h} \cup \{\mathbf{e}^{(t)}\}|/|\Theta_\text{trgt}| \in [0.01, 2.0]$ **(a)** split MNIST performance for CL1 stays high even for compression ratios $\approx 1\%$. **(b)** PermutedMNIST-10 accuracies degrade gracefully when compression ratios decline to $1\%$. Notably, for both benchmarks, performance remained stable across a large pool of hypernetwork configurations.

the compressive regime. Note that the discussed compression ratio compares the amount of trainable parameters in the hypernetwork to its output size, i.e. the parameters of the target network.

**Small capacity target networks for the permuted MNIST benchmark.** Swaroop et al. (2018) argue for using only small capacity target networks for this benchmark. Specifically, they propose to use hidden layer sizes $[100, 100]$. Again, we replicated the setup of van de Ven & Tolias (2019) wherever applicable, except for the now smaller hidden layer sizes of $[100, 100]$ in the target network. We use a fully-connected chunked hypernetwork with chunk embeddings $\mathbf{c}$ having size 12, hidden layers having size 100, 75, 50 and an output size of 2000, resulting in a total number of hypernetwork weights of 122,459 (including $10 \times 64$ task embedding weights) compared to 122,700 weights that are generated for the target network. $\beta_\text{output}$ is set to 0.05. The experiments performed here correspond to CL1.

We achieve an average accuracy of $93.91 \pm 0.04$ for PermutedMNIST-10 after having trained on all tasks. In general, we saw that the hypernetwork training can benefit from noise injection. For instance, when training with soft-targets (i.e., we modified the 1-hot target to be 0.95 for the correct class and $\frac{1-0.95}{\#\text{classes}-1}$ for the remaining classes), we could improve the average accuracy to $94.24 \pm 0.03$.

We also checked the challenging PermutedMNIST-50 benchmark with this small target network as previously investigated by Ritter et al. (2018). Therefore, we slightly adapted the above setup by using a hypernetwork with hidden layer sizes $[100, 100]$ and a regularization strength of $\beta_\text{output} = 0.1$. This hypernetwork is slightly bigger than the corresponding target network $\frac{|\Theta_\text{h} \cup \{\mathbf{e}^{(t)}\}|}{|\Theta_\text{trgt}|} = 1.37$. With this configuration, we obtain an average accuracy of $90.91 \pm 0.07$.

**Comparison to HAT.** Serra et al. (2018) proposed the hard attention to the task (HAT) algorithm, a strong CL1 method which relies on learning a per-task, per-neuron mask. Since the masks are pushed to become binary, HAT can be viewed as an algorithm for allocating subnetworks (or modules) within the target network, which become specialized to solve a given task. Thus, the method is similar to ours in the sense that the computation of the target network is task-dependent, but different in spirit, as it relies on network modularity.

In HAT, task identity is assumed to be provided, so that the appropriate mask can be picked during inference (scenario CL1). HAT requires explicitly storing a neural mask for each task, whose size

scales with the number of neurons in the target network. In contrast, our method allows solving tasks in a compressive regime. Thanks to the hypernetwork, whose input dimension can be freely chosen, only a low-dimensional embedding needs to be stored per task (cf. Fig. 4), and through chunking it is possible to learn to parameterize large target models with a small number of plastic weights (cf. Fig. 3b).

Here, we compare our task-conditioned hypernetworks to HAT on the permuted MNIST benchmarks ($T = 10$ and $T = 100$), cf. Table 6. For large target networks, both methods perform strongly, reaching comparable final task-averaged accuracies. For small target network sizes, task-conditioned hypernetworks perform better, the difference becoming more apparent on PermutedMNIST-100.

We note that the two algorithms use different training setups. In particular, HAT uses 200 epochs (batch size set to 64) and applies a learning rate scheduler that acts on a held out validation set. Furthermore, HAT uses differently tuned forgetting hyperparameters when target network sizes change. This is important to control for the target network capacity used per task and assumes knowledge of the (number of) tasks at hand. Using the code freely made available by the authors, we were able to rerun HAT for our target network size and longer task sequences. Here, we used the setup provided by the author's code for HAT-Large for PermutedMNIST-10 and PermutedMNIST-100. To draw a fairer comparison, when changing our usual target network size to match the ones reported in Serra et al. (2018), we trained for 50 epochs per task (no training loss improvements afterwards observed) and also changed the batch size to 64 but did not changed our training scheme otherwise; in particular, we did not use a learning rate scheduler.

Table 6: Comparison of HNET and HAT, Serra et al. (2018). Task-averaged test accuracy on the PermutedMNIST experiment with $T = 10$ and $T = 100$ tasks ('P10', 'P100') with three different target network sizes, i.e., three fully connected neural networks with hidden layer sizes of $(100, 100)$ or $(500, 500)$ or $(2000, 2000)$ are shown. For these architectures, a single accuracy was reported by Serra et al. (2018) without statistics provided. We reran HAT for PermutedMNIST-100 with code provided at `https://github.com/joansj/hat`, and for PermutedMNIST-10 with hidden layer size $(1000, 1000)$ to match our setup. HAT and HNET perform similarly on large target networks for PermutedMNIST-10, while HNET is able to achieve larger performances with smaller target networks as well as for long task sequences.

|  | **HAT** | **HNET** |
| --- | --- | --- |
| P10-`100,100` | 91.6 | $95.92 \pm 0.02$ |
| P10-`500,500` | 97.4 | $97.35 \pm 0.02$ |
| P10-`2000,2000` | 98.6 | $98.06 \pm 0.02$ |
| P10-`1000,1000` | $97.67 \pm 0.02$ | $97.56 \pm 0.02$ |
| P100-`1000,1000` | $86.04 \pm 0.26$ | $94.98 \pm 0.07$ |

**Efficient PermutedMNIST-250 experiments with a stochastic regularizer on subsets of previous tasks.** An apparent drawback of Eq. 2 is that the runtime complexity of the regularizer grows linearly with the number of tasks. To overcome this obstacle, we show here that it is sufficient to consider a small random subset of previous tasks.

In particular, we consider the PermutedMNIST-250 benchmark (250 tasks) on `CL1` using the hyper-parameter setup from our PermutedMNIST-100 experiments except for a hypernetwork output size of 12000 (to adjust to the bigger multi-head target network) and a regularization strength $\beta_{\text{output}} = 0.1$. Per training iteration, we choose maximally 32 random previous tasks to estimate the regularizer from Eq. 2. With this setup, we achieve a final average accuracy of $94.19 \pm 0.16$ (compared to an average during accuracy (i.e., the accuracies achieved right after training on the corresponding task) of $95.54 \pm 0.05$). All results are across 5 random seeds. These results indicate that a full evaluation of the regularizer at every training iteration is not necessary such that the linear runtime complexity can be cropped to a constant one.

**Combining hypernetwork output regularizers with weight importance.** Our hypernetwork regularizer pulls uniformly in every direction, but it is possible to introduce anisotropy using an EWC-like approach (Kirkpatrick et al., 2017). Instead of weighting parameters, hypernetwork outputs can be weighted. This would allow for a more flexible regularizer, at the expense of additional storage.

**Task inference through predictive entropy (HNET+ENT).** In this setup, we rely on the capability of neural networks to separate in- from out-of-distribution data. Although this is a difficult research problem on its own, for continual learning, we face a potentially simpler problem, namely to detect and distinguish between the tasks our network was trained on. We here take the first minimal step exploiting this insight and compare the predictive uncertainty, as quantified by output distribution entropy, of the different models given an input. Hence, during test time we iterate over all embeddings and therefore the models our metamodel can generate and compare the predictive entropies which results in making a prediction with the model of lowest entropy. For future work, we wish to explore the possibility of improving our predictive uncertainty by taking parameter uncertainty into account through the generation of approximate, task-specific weight posterior distributions.

**Learning without task boundaries with hypernetworks.** An interesting problem we did not address in this paper is that of learning without task boundaries. For most CL methods, it is crucial to know when learning one task ends and training of a new tasks begins. This is no exception for the methods introduced in this paper. However, this is not necessarily a realistic or desirable assumption; often, one desires to learn in an online fashion without task boundary supervision, which is particularly relevant for reinforcement learning scenarios where incoming data distributions are frequently subject to change (Rolnick et al., 2018). At least for discrete changes, with our hypernetwork setup, this boils down to a detection mechanism that activates the saving of the current model, i.e., the embedding $\mathbf{e}^{(T)}$, and its storage to the collection of embeddings $\{\mathbf{e}^{(t)}\}$. We leave the integration of our model with such a hypernetwork-specific switching detection mechanism for future work. Interestingly, our task-conditioned hypernetworks would fit very well with methods that rely on fast remembering (a recently proposed approach which appeared in parallel to our paper, He et al., 2019).

# E UNIVERSAL FUNCTION APPROXIMATION WITH CHUNKED NEURAL NETWORKS

**Proposition 1.** *Given a compact subset $K \subset \mathbb{R}^m$ and a continuous function on $K$ i.e. $f \in C(K)$, more specifically, $f : K \to \mathbb{R}^n$ with $n = r \cdot N_C$. Now $\forall \epsilon > 0$, there exists a chunked neural network $f_{\mathrm{h}}^{\mathbf{c}} : \mathbb{R}^m \times \mathcal{C} \to \mathbb{R}^r$ with parameters $\Theta_{\mathrm{h}}$, discrete set $\mathcal{C} = \{\mathbf{c}_1, \ldots, \mathbf{c}_{N_C}\}$ and $\mathbf{c}_i \in \mathbb{R}^s$ such that $|\bar{f}_{\mathrm{h}}^{\mathbf{c}}(\mathbf{x}) - f(\mathbf{x})| < \epsilon, \quad \forall \mathbf{x} \in K$ and with $\bar{f}_{\mathrm{h}}^{\mathbf{c}}(\mathbf{x}) = [f_{\mathrm{h}}^{\mathbf{c}}(\mathbf{x}, \mathbf{c}_1), \ldots, f_{\mathrm{h}}^{\mathbf{c}}(\mathbf{x}, \mathbf{c}_{N_C})]$.*

For the following proof, we assume the existence of one form of the universal approximation theorem (UAT) for neural networks (Leshno & Schocken, 1993; Hanin, 2017). Note that we will not restrict ourselves to a specific architecture, nonlinearity, input or output dimension. Any neural network that is proven to be a *universal function approximator* is sufficient.

*Proof.* Given any $\epsilon > 0$, we assume the existence of a neural network $f_{\mathrm{h}} : \mathbb{R}^m \to \mathbb{R}^n$ that approximates function $f$ on $K$:

$$|f_{\mathrm{h}}(\mathbf{x}) - f(\mathbf{x})| < \frac{\epsilon}{2}, \quad \forall x \in K. \tag{10}$$

We will in the following show that we can always find a chunked neural network $f_{\mathrm{h}}^{\mathbf{c}} : \mathbb{R}^m \times \mathcal{C} \to \mathbb{R}^r$ approximating the neural network $f_{\mathrm{h}}$ on $K$ and conclude with the triangle inequality

$$|\bar{f}_{\mathrm{h}}^{\mathbf{c}}(\mathbf{x}) - f(\mathbf{x})| \le |\bar{f}_{\mathrm{h}}^{\mathbf{c}}(\mathbf{x}) - f_{\mathrm{h}}(\mathbf{x})| + |f_{\mathrm{h}}(\mathbf{x}) - f(\mathbf{x})| < \epsilon, \quad \forall x \in K. \tag{11}$$

Indeed, given the neural network $f_{\mathrm{h}}$ such that (10) holds true, we construct

$$\hat{f}_{\mathrm{h}}(\mathbf{x}, \mathbf{c}) = \begin{cases} f_{\mathrm{h}}^{\mathbf{c}_i}(\mathbf{x}) & \mathbf{c} = \mathbf{c}_i \\ 0 & \text{else} \end{cases} \tag{12}$$

by splitting the *full* neural network $f_{\mathrm{h}}(\mathbf{x}) = [f_{\mathrm{h}}^{\mathbf{c}_1}(\mathbf{x}), f_{\mathrm{h}}^{\mathbf{c}_2}(\mathbf{x}), \ldots, f_{\mathrm{h}}^{\mathbf{c}_{N_C}}(\mathbf{x})]$ with $\hat{f}_{\mathrm{h}} : \mathbb{R}^m \times \mathcal{C} \to \mathbb{R}^r$.

Note that $\hat{f}_{\mathrm{h}}$ is continuous on $\mathbb{R}^m \times \mathcal{C}$ with the product topology composed of the topology on $\mathbb{R}^m$ induced by the metric $|\cdot - \cdot| : \mathbb{R}^m \times \mathbb{R}^m \to \mathbb{R}$ and the discrete topology on $\mathcal{C}$. Now we can make use

of the UAT again: Given the compact $K \subset \mathbb{R}^n$, the discrete set $\mathcal{C} = \{\mathbf{c}_1, \dots, \mathbf{c}_{N_C}\}$ and any $\frac{\epsilon}{2N_C} > 0$, there exists a neural network function $f_\mathrm{h}^{\mathbf{c}} : \mathbb{R}^m \times \mathbb{R}^s \to \mathbb{R}^r$ such that

$$|f_\mathrm{h}^{\mathbf{c}}(\mathbf{x}, \mathbf{c}) - \hat{f}_\mathrm{h}(\mathbf{x}, \mathbf{c})| < \frac{\epsilon}{2N_C}, \quad \forall \mathbf{x} \in K, \forall \mathbf{c} \in \mathcal{C}. \tag{13}$$

It follows that

$$\sum_i |f_\mathrm{h}^{\mathbf{c}}(\mathbf{x}, \mathbf{c}_i) - \hat{f}_\mathrm{h}(\mathbf{x}, \mathbf{c}_i)| < \sum_i \frac{\epsilon}{2N_C} = \frac{\epsilon}{2}, \quad \forall \mathbf{x} \in K, \tag{14}$$

which is equivalent to

$$\left| \begin{bmatrix} f_\mathrm{h}^{\mathbf{c}}(\mathbf{x}, \mathbf{c_1}) \\ \vdots \\ f_\mathrm{h}^c(\mathbf{x}, \mathbf{c_{N_C}}) \end{bmatrix} - \begin{bmatrix} \hat{f}_\mathrm{h}(\mathbf{x}, \mathbf{c_1}) \\ \vdots \\ \hat{f}_\mathrm{h}(\mathbf{x}, \mathbf{c_{N_C}}) \end{bmatrix} \right| = |\bar{f}_\mathrm{h}^{\mathbf{c}}(\mathbf{x}) - f_\mathrm{h}(\mathbf{x})| < \frac{\epsilon}{2}, \quad \forall \mathbf{x} \in K. \tag{15}$$

We have shown (11) which concludes the proof. $\qquad\square$

Note that we did not specify the number of chunks $N_C$, $r$ or the dimension $s$ of the embeddings $\mathbf{c}_i$. Despite this theoretical result, we emphasize that we are not aware of a constructive procedure to define a chunked hypernetwork that comes with a useful bound on the achievable performance and/or compression rate. We evaluate such aspects empirically in our experimental section.

## F    QUALITATIVE ANALYSES OF HYPERNETWORK-PROTECTED REPLAY MODELS

**a**                                                                                    **b**

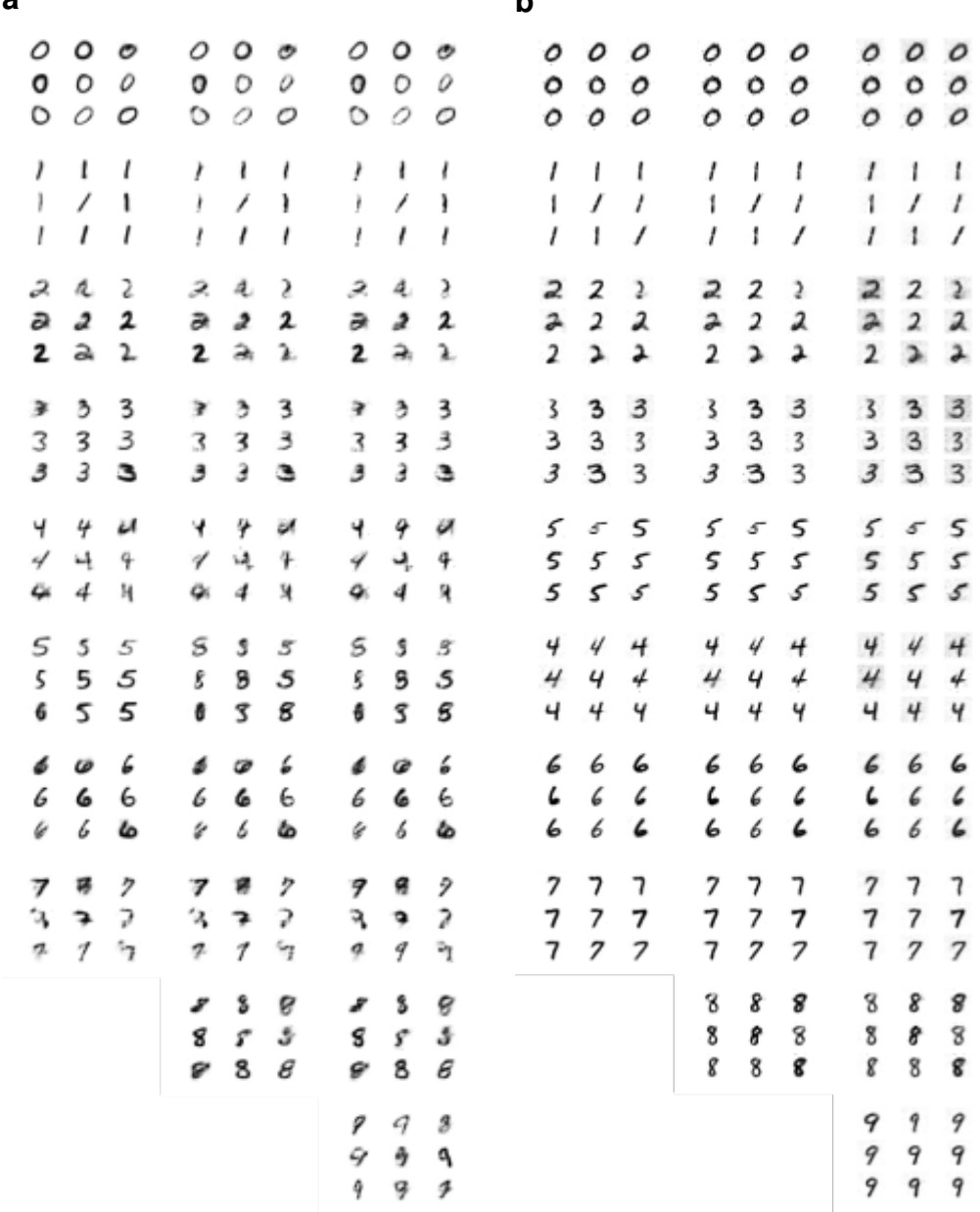

Figure A8: **Image samples from hypernetwork-protected replay models.** The left column of both of the subfigures display images directly after training the replay model on the corresponding class, compared to the right column(s) where samples are obtained after training on eights and nines i.e. all classes. **(a)** Image samples from a class-incrementally trained VAE. Here the exact same training configuration to obtain results for split MNIST with the HNET+R setup are used, see Appendix C. **(b)** Image samples from a class-incrementally trained GAN. For the training configurations, see Appendix B. In both cases the weights of the generative part, i.e., the decoder or the generator, are produced and protected by a hypernetwork.

