# OpenReview forum: "Continual learning with hypernetworks"
_ICLR.cc/2020/Conference — Accept (Spotlight)_

### Official Review · AnonReviewer1 · 2019-10-22
**Official Blind Review #1**

**Rating:** 6

**Review:**

Paper 1872
Paper proposes a method for CL. The method is based on hypernetworks. These networks are a metamodel, which produce the parameters (from a task-conditioned embedding) which will be used in the main network. Preventing forgetting in the main network is now, replaced by preventing forgetting in the hypernetwork. This is done by imposing a regularization on the hypernetwork outcome, imposing that the generated weights should be similar for previous tasks (similar to Li & Hoiem who impose this on the network outputs). In addition, the paper proposes chunking, which refers to using an additional set of chunk, embeddings which are shared for all tasks, which allow compressing the hypernetwork. Furthermore, they propose an extension that allows for image replay (this is not an easy extension and an impressive contribution on itself, but maybe confusing for the current paper).

CONCLUSION
Overall, I like the idea of the paper and it is well explained. However, I found that the experiments of the paper where not well designed to verify the main contribution (hypernetworks), nor where they compared to the most relevant methods. I am borderline with this paper, and recommend borderline accept (borderline not being an option).

QUESTIONS
1. I think the motivation of why hypernetworks are expected to have less forgetting (than addressing forgetting directly in a network) should be discussed early in the paper.

2.I do not understand why the training is performed in two steps. First computing a candidate Delta THETA_H and then o ptimizing Eq 2. Why not directly optimizing Eq 2, replacing the second factor with|| f_h(e^t, THETA^*_h)-f_h(e^t, THETA_h) ||. This is how this regularization is normally applied (e.g. Li & Hoiem). If the authors insist in using Eq 2, I would like to see it compared with the proposed version.

3. The experiments should show that hypernets better address CL then addressing this directly in the network (and preferably provide reasons for this). Comparison with the closest methods like HAT and PackNet should be included. Especially, HAT is interesting since it is also based on an embedding.

4. Also, more experiments on CIFAR would be welcome. The MNIST variations already provide very high accuracies. For CIFAR-10/100 groups of 20 classes are added in 5 steps ? Scenario CL3 would be interesting for CIFAR as well.

5. I would like to see more analysis and results for the chunking. (As said before the replay is also a nice addition, but it seems an add-on of the main-text, shrinking the space to analyze the main contributions of the paper in the experiments.)
I guess HNET+ENT for CL1 scenario does not use ENT and is just HNET?


**Experience Assessment:**

I have published one or two papers in this area.

**Review Assessment: Checking Correctness Of Derivations And Theory:**

I assessed the sensibility of the derivations and theory.

**Review Assessment: Checking Correctness Of Experiments:**

I assessed the sensibility of the experiments.

**Review Assessment: Thoroughness In Paper Reading:**

I read the paper at least twice and used my best judgement in assessing the paper.

---

> ### Author Response · Authors · 2019-11-13
> **Response to AnonReviewer1 (Part 1/2)**
>
> We thank the reviewer for his positive feedback and constructive comments. We ran additional experiments triggered by the questions that were raised in the assessment of our manuscript and modified it accordingly, as detailed below.
>
> Q: I think the motivation of why hypernetworks are expected to have less forgetting (than addressing forgetting directly in a network) should be discussed early in the paper.
>
> We have taken up the reviewer's suggestion and now present and motivate task-conditioned hypernetworks already in the second paragraph of the introduction.
>
> Q: Parameter lookahead on the regularization term.
>
> The reviewer is correct in that the lookahead trick (used also for example by Benjamin et al., 2018) is a rather minor addition to our method. This is now stressed in the manuscript (paragraph "Continual learning with hypernetwork output regularization"). The exploratory hyperparameter sweeps that we conducted on MNIST revealed that the inclusion of the parameter lookahead brings a modest, though significant, task-averaged accuracy increase of about 2\% on PermutedMNIST-100.
>
> Q: I found that the experiments of the paper where not well designed to verify the main contribution (hypernetworks), nor where they compared to the most relevant methods. Comparison with the closest methods like HAT and PackNet should be included. Especially, HAT is interesting since it is also based on an embedding.
>
> We kindly disagree with the reviewer's assessment on our choice of benchmark methods. We chose DGR since it is a strong baseline across the three CL scenarios and because it is the natural and direct comparison for hypernetwork-protected replay. As a second comparison, we chose weight regularization methods (EWC and SI) as this technique is closely related to ours, serving as a baseline that illustrates the advantages of performing weight regularization at the meta-level. Indeed, one interpretation of our regularizer is to see it as naive (uniform) EWC at the meta-level. This view suggests using non-uniform weight importances at the meta-level, determined e.g. using Laplace's approximation, an avenue that we leave open for future work.
>
> We do agree with the reviewer that masking methods are an interesting approach to CL, related to ours in the sense that masks are typically specified and stored on a task-by-task basis. Using the code that was made available by the authors of the HAT paper, we were able to compare our methods on the CL1 permuted MNIST benchmark, for T=10 and T=100. While the methods perform similarly for large target models, we found that task-conditioned hypernetworks display better performances when using smaller target networks, with a more pronounced advantage on long task sequences (T=100). We have included these new results on Appendix D (cf. Table 3).
>
> Performance comparisons aside, we would like to point out what we believe are a few important advances that our paper introduces. First, we note that the term "embedding" is used with a different meaning by the HAT paper authors, as it corresponds to a neuron mask in that method. Therefore, a potentially large vector has to be additionally stored per task in HAT. For example, on a ResNet-32 there are ~300k neurons and only ~460k weights, so the dimension of a neuron mask is not far from that of the network itself. Second, in our approach, masks are not explicitly stored aside but learned through a metamodel, which enables compression and generalization in embedding space (cf. Fig. 4). Finally, both HAT and PackNet are specifically designed and tested only for CL1 problems, and an extension of those methods to CL2, CL3 and replay models will likely require considerable further investigation.
>
> Q: Additional experiments.
>
> We have modified the paper to include a new set of extensive results on the CL2/CL3 PermutedMNIST-100 benchmark (Fig. A4). Such study of long task sequences beyond CL1 is the first of the kind, to the best of our knowledge. We found that hypernetwork-protected replay, together with a task inference network (HNET+TIR), markedly and significantly outperforms both EWC/SI and standalone generative replay. We believe that these new strong results further motivate the potential benefits of a divide-and-conquer approach, where multiple complementary systems act in tandem to enable CL. We leave to future work the investigation of hypernetwork-protected replay models for CL2/3 on more challenging datasets such as CIFAR-10/100.
>
>
> Q: For CIFAR-10/100 groups of 20 classes are added in 5 steps?
>
> We consider the benchmark introduced by Zenke et al (2018), where the entire CIFAR-10 dataset is first solved, followed by five sets of ten CIFAR-100 classes. We have now clarified the manuscript on this point (beginning of "split CIFAR-10/100 benchmark" paragraph).
>
> (Continued: please see part 2/2)

---

> > ### Author Response · Authors · 2019-11-13
> > **Response to AnonReviewer1 (Part 2/2)**
> >
> > (Continuation of Part 1/2)
> >
> > Q: More analysis and results for the chunking.
> >
> > On a new Appendix section, see Fig. A7, we now report a robustness analysis that we have carried out on both split MNIST and permuted MNIST for a large number of hypernetwork models with varying chunk sizes. Our analysis shows that performance is stable across a broad range of compression levels and hypernetwork architectures. We hope that these findings inspire more extensive future work on hypernetwork architecture search, which will likely impact a growing number of hypernetwork-based models, the present one included.
> >
> > Q: I guess HNET+ENT for CL1 scenario is just HNET?
> >
> > Correct. This has been clarified in the manuscript (Table 1 caption).

---

### Official Review · AnonReviewer3 · 2019-10-23
**Official Blind Review #3**

**Rating:** 8

**Review:**

Review of “Continual learning with hypernetworks”

This paper investigates the use of conditioning “Hypernetworks” (networks where weights are compressed via an embedding) for continual learning. They use “chunked” version of the hypernetwork (used in Ha2017, Pawlowski2017) to learn task-specific embeddings to generate (or map) tasks to weights-space of the target network.

There is a list of noteworthy contributions of this work:

1) They demonstrate that their approach achieves SOTA on various (well-chosen) standard CL benchmarks (notably P-MNIST for CL, Split MNIST) and also does reasonably well on Split CIFAR-10/100 benchmark. The authors have also spent some effort to replicate previous work so that their results can be compared (and more importantly analyzed) fairly to the literature, and I want to see more of this in current ML papers. (one note is that the results for CIFAR-10/100 is in the Appendix, but I think if the paper gets accepted, let's bring it back to the main text and use 9 pages, since the results for CIFAR10/100 are substantial).

2) In addition to demonstrating good results on standard CL benchmarks, they also conduct analysis of task-conditioned hypernetworks with experiments involving long task sequences to show that they have very large capacities to retain large memories. They provide a treatment (also with visualization) into the structure of low-dim task embeddings to show potential for transfer learning.

3) The authors will release code to reproduce all experiments, which I think is important to push the field forward. Future work can not only reproduce this work, but also the cited works.

The work seems to be well written, and the motivation of using hypernetworks as a natural solution to avoid catastrophic loss is clearly described. Overall, I think this work is worthy of acceptance and should encourage more investigation into hypernetworks for CL and transfer learning going forward in the community.


**Experience Assessment:**

I have published one or two papers in this area.

**Review Assessment: Checking Correctness Of Derivations And Theory:**

I did not assess the derivations or theory.

**Review Assessment: Checking Correctness Of Experiments:**

I assessed the sensibility of the experiments.

**Review Assessment: Thoroughness In Paper Reading:**

I read the paper at least twice and used my best judgement in assessing the paper.

---

> ### Author Response · Authors · 2019-11-13
> **Response to AnonReviewer3**
>
> We thank the reviewer for his encouraging feedback and appreciation of our efforts, that we very much enjoyed reading.
>
> We are happy to follow the proposed suggestion and move the additional split CIFAR-10/100 experiments to the main text if the paper gets accepted.

---

### Official Review · AnonReviewer2 · 2019-10-23
**Official Blind Review #2**

**Rating:** 6

**Review:**

This paper proposes to use hypernetwork to prevent catastrophic forgetting. In deep learning, the information of the samples are converted to parameters during the training process, however, future training process could interfere with the information from the previous tasks. One of the method to prevent forgetting is to use reheasal, which retrains the network with previous data. The mechanism of this work is to store the previous samples as a trained point in the parameter space, so that a set of points in the original space is stored and thus rehearsed as one point in the parameter space, this saves both the memory and computation.

I give a weak accept of this paper due to the following reasons:
Pros:
- The idea of converting a set of data points to one point and rehearse at a meta level is a smart and novel idea.
- It shows significant improvement compared to baseline methods, especially for split CIFAR experiments.
- The Appendix contains a fair amount of details and additional experiments on generative models.

Cons:
- This works assumes a task incremental setting, during training process task is received one by one, within each task we could assume i.i.d shuffling of the data. During testing, the task boundary is optional. Although this setting has been taken by many other works in this field, it is also criticised that availability of task boundary is an unrealistic setting. A more realistic setting would be to continually learn with a continuous non-stationary stream of data, which indicates there's no split of train / test phase. Thus a general continual learning method should not require task boundary, which would be problematic for this work as it depends on task conditioning.
- For the rehearsal objective in 2, L2 penalty is used. This could be a problem as minimizing the L2 distance in the parameter space does not necessarily minimize the task loss.

Questions I have that needs clarification:
- Chunking: Are the chunking parameters shared/updated across tasks?

**Experience Assessment:**

I have published one or two papers in this area.

**Review Assessment: Checking Correctness Of Derivations And Theory:**

I assessed the sensibility of the derivations and theory.

**Review Assessment: Checking Correctness Of Experiments:**

I assessed the sensibility of the experiments.

**Review Assessment: Thoroughness In Paper Reading:**

I read the paper at least twice and used my best judgement in assessing the paper.

---

> ### Author Response · Authors · 2019-11-13
> **Response to AnonReviewer2**
>
> We thank the reviewer for his overall positive feedback and appreciation of our work. We reply individually to each raised point below.
>
> Q: A more realistic setting would be to continually learn with a continuous non-stationary stream of data, which indicates there's no split of train / test phase. Thus a general continual learning method should not require task boundary, which would be problematic for this work as it depends on task conditioning.
>
> We fully agree with the reviewer that learning subject to a continuously evolving data distribution is a fundamental research problem, distinct from the one considered in our work. Notwithstanding, in many situations of practical interest it seems appropriate to assume that the data are generated conditioned on a switching, discrete task (or context) latent variable. For supervised learning, often, task identity is naturally available during training (e.g., on a class-incremental learning scenario). In certain cases, however, in line with the reviewer's point, such task variable might not be explicitly available to the supervisor (see, e.g., He et al., 2019). We believe that task-conditioned hypernetworks are a promising approach on such a scenario, provided that an additional task boundary detection mechanism can be integrated into the system. We leave the investigation of such a complementary mechanism to future work.
>
> Q: On the L2 penalty.
>
> The reviewer is right, closeness in parameter space measured by the L2 norm does not necessarily imply closeness measured in the actual output loss. Interestingly, the L2 loss in parameter space can be related in certain conditions to a distance in function space (Benjamin et al., 2018). A study of alternative penalties, for example using the Fisher metric, or the KL divergence in a variational inference framework, is an interesting direction for improving the present method.
>
> Q: Are the chunking parameters shared/updated across tasks?
>
> Correct. A single set of chunk embeddings is shared and updated across tasks, and treated like other parameters in $\Theta_h$.

---

### Author Response · Authors · 2019-11-13
**Collective response to reviewers**

We are very grateful to all three reviewers for the time taken in carefully assessing our work and for the overall positive feedback, that we found very encouraging.

We have added new results including a study of the PermutedMNIST-100 CL2/3 benchmark and improved the clarity of the manuscript. We also provide new analyses on chunking and a comparison on PermutedMNIST-10/100 to a masking CL method known as HAT, following AnonReviewer1's suggestions. We believe that this has significantly improved the paper and we sincerely hope that AnonReviewer1 and AnonReviewer2 would consider increasing their rating from a weak accept to accept.

We have placed the PermutedMNIST-100 CL2/3 results on the Appendix. Following AnonReviewer3's suggestion to increase the paper length to nine pages upon acceptance, and if the reviewers find such change appropriate, we could use the additional space and move Fig. A4 to the main text.

---

### Decision · Program_Chairs · 2019-12-19

**Decision:**

Accept (Spotlight)

**Comment:**

This paper proposes to use hypernetwork to prevent catastrophic forgetting. Overall, the paper is well-written, well-motivated, and the idea is novel. Experimentally, the proposed approach achieves SOTA on various (well-chosen) standard CL benchmarks (notably P-MNIST for CL, Split MNIST) and also does reasonably well on Split CIFAR-10/100 benchmark. The authors are suggested to investigate alternative penalties in the rehearsal objective, and also add comparison with methods like HAT and PackNet.